# Inheritable cell-states shape drug-persister correlations and population dynamics in cancer cells

Anton Iyer[1], Adrian Alva[1], Adrián E. Granada[2], Shaon Chakrabarti (iD)[1]*

**1** Simons Centre for the Study of Living Machines, National Centre for Biological Sciences, Bangalore, India, **2** Charité Comprehensive Cancer Center, Berlin, Germany

* shaon@ncbs.res.in

## Abstract

Drug tolerant persisters (DTPs) drive cancer therapy resistance by temporarily evading drug action, allowing multiple routes to eventual permanent resistance. Despite clear evidence for DTPs, the timing of their emergence, proliferative nature, and how their population dynamics arise from measured single-cell kinetics remain poorly understood. Here we use time-lapse microscopy data from two cancer cell lines, integrating single-cell and population measurements, to develop a quantitative description of drug persistence. Contrary to the expectation that increasing levels of genotoxic stress should lead to slower times to division and faster times to death, we observe minor changes in the single-cell intermitotic and death time distributions upon increasing cisplatin concentration. Yet, population decay rates increase 3-fold, suggesting a surprising independence of the overall dynamics from the measured birth and death rates. To explain this phenomenon, we argue that the observed lineage correlations and concentration-dependent decay rates imply cell-state dependent fate choices made *both* pre and post-cisplatin as opposed to just post-drug birth/death rate-based competitive fate choices. We demonstrate that these cell-states, present in the pre-drug ancestors of DTP and sensitive cells, exhibit no difference in cycling speed and are inherited across at least 2-3 cellular generations. Post-drug survival versus death fates are decided with high probability by these pre-existing cell-states, but get modulated to some extent by the drug, leading to a drug concentration dependent state-fate map. A stochastic model implementing these rules simultaneously recapitulates the observed decay rates and cell-fate lineage correlations. The model also demonstrates how the use of barcode diversity change before and after drug might lead to misleading interpretations of the timing of persister fate decisions. Our results provide a conceptual framework for quantifying pre versus post-drug contributions to cell fate, without requiring knowledge of the underlying molecular architecture of the heterogeneous cell states.

**Data availability statement:** All code will be available on https://github.com/Shaonlab.

**Funding:** S.C. acknowledges intramural funds, and a regular salary from the National Center for Biological Sciences–Tata Institute of Fundamental Research (NCBS-TIFR) via Department of Atomic Energy, Government of India, Project Identification No. RTI 4006. The funder had no role in study design, data collection and analysis, decision to publish, or preparation of the manuscript.

**Competing interests:** The authors have declared that no competing interests exist.

## Author summary

It is now well understood that evasion of drug effects on short time-scales by cancer cells (drug tolerant persisters or DTPs) originates from cell fate outcomes driven by non-genetic heterogeneity. However, the timing of these fate decisions is currently debated. Here we combine two scales of cancer cell proliferation – single-cell and population dynamics, to argue that fate decisions must occur both before and post drug administration. This conclusion emerges from developing a series of models of plausible cell fate decision processes, to best explain a set of puzzling observations in microscopy datasets: (1) though the population dynamics post cisplatin varies by 3-fold between high and low drug concentrations, there is no detectable change in the single-cell kinetics, and (2) end fate is strongly correlated between cells closer in lineage distance. The final model we develop quantitatively explains all these observations, and provides a conceptual framework for quantifying the contribution of pre versus post-drug cell fate decisions. Our results also demonstrate that measuring drug-induced changes in barcode diversity can lead to misleading interpretations on persister timing. Our conclusions do not depend on identifying the molecular origins of drug tolerance, providing a general framework for investigating the timing of persister development.

## Introduction

Non-genetic heterogeneity amongst isogenic cells gives rise to drug-tolerant persisters or "DTP"s – cells that temporarily survive drug treatment, often providing a reservoir for future genetic changes driving permanent resistance [1–8]. Originally discovered in the context of bacterial-cell survival in the presence of antibiotics [9], the existence of persisters was later suggested for cancer as well, where a fractional-killing effect was demonstrated to limit curability in leukemia [10]. A large body of literature has since accumulated, suggesting interesting parallels between the phenomenon of persistence in bacterial and cancer cells, including a characteristic bi-phasic exponential decay in the population dynamics after drug administration and the origin of persisters from a small fraction of quiescent or slow-cycling cells in drug-naive conditions [8,11–14]. Furthermore, there seems to be no unique non-genetic cell-state that gives rise to persistence either in bacteria or in cancer cells – a wide range of regulatory networks such as metabolic, extra-cellular matrix and EMT pathways are likely involved depending on the specifics of the cell-type, the drug being administered and cellular interactions with the micro-environment.

While there have been a lot of recent advances in characterizing DTP's and their origins, many questions remain unresolved and debated, particularly in the context of cancer drug tolerance [15]. For instance, the timing of cellular decisions leading to DTPs remains unclear. Based on end-fate (survival or death) correlations on lineages, some previous studies concluded that heterogeneous cell states existing *before* drug administration ("pre-DTPs") predetermine cell fate outcomes post drug treatment [2,16–18]. Conceptually similar conclusions were later drawn from elegant barcoding techniques that additionally gave insights into the genes that might determine these pre-DTP states [19–21], giving rise to the notion of "memory genes" [19,21,22]. It was also demonstrated using theoretical models of transcriptional bursting that it is indeed possible to generate rare cell-states where fluctuations in memory gene expression persist for a few generations, leading to cell survival in the face of drug treatment [23]. However, other studies have argued that fate decisions are drug-induced by combining mathematical modeling of cancer population dynamics, modified

Luria-Delbrück fluctuation tests and barcoding approaches [7,24,25]. Similarly, the question of whether persister cells always arise from lineages with intrinsically slower division kinetics remains debated. Original studies in bacteria attributed persisters to the presence of slowly cycling sub-populations in drug naive conditions [12,13]. Most studies in cancer cells have also suggested the same phenomenon, resulting in the terms "cancer persisters" and "quiescence/dormant" being used almost interchangeably [26,27]. These studies suggest the presence of a fitness cost to non-genetic persistence in the same vein as is often ascribed to genetic resistance mechanisms [28]. However, emerging evidence suggests that the ancestors of cancer persisters before drug treatment might also be proficient in cell-division, therefore challenging the wide-spread notion that drug tolerant persisters necessarily emerge from quiescent subpopulations of cancer cells [20,29,30].

In this study we explore the nature, timing and kinetics of these cell fate decisions by investigating how the observed population dynamics of cancer cells quantitatively emerges from the measured single-cell kinetics. Integrating single-cell and population dynamics measurements from time-lapse microscopy datasets, we first demonstrate the surprising result that increasing cisplatin concentration does not affect cellular birth or death rates, unlike what is widely assumed [31–36]. As a consequence, the ubiquitous exponential growth models which assume stochastic competition of cell fates (birth-death process and age-structured population models), fail to explain experimental data from two cisplatin-treated cell types, HCT116 and U2OS. Instead, we argue based on observed lineage correlations that pre-existing cell states largely determine fate decisions, and are inherited across multiple cellular generations before drug addition. However, some degree of fate determination must also occur during the time of drug administration, which is necessary to explain drug concentration-dependent effects on the population dynamics. We quantify these arguments with a simple model that allows disentangling the pre versus post-drug contributions to cell fate. Furthermore, we demonstrate that cells in states that are primed for death or survival do not have fitness differences, and are both inherited across at least 2-3 generations before the drug is administered. Finally, we show how early fate decisions are consistent with recent barcoding experiments where no change in barcode diversity was observed after drug administration. Our arguments are based on general principles independent of the molecular details of the underlying cellular states, providing a powerful approach to deciphering the dynamics of drug persistence in cancer.

## Methods

### Mixture models to investigate the presence of multiple cellular populations before cisplatin, with different cycling kinetics

While full lineages were tracked in the HCT116 dataset [18], only one cell per sister-pair was tracked in the U2OS dataset [37]. Hence for the latter, we resorted to using a mixture model to check whether multiple subpopulations with distinct proliferation rates could be identified from the pre-cisplatin cells. Since previous work has identified the Exponentially Modified Gaussian (EMG) as a good model of the IMT distribution [18], we used a mixture of EMGs for this purpose. The EMG is a convolution of an exponential and Gaussian distributions, the former parameterized by $\lambda$ and the latter by $\mu$ and $\sigma$. We used a mixture of two EMGs to fit the IMT distribution data and compared with the fit to a single EMG using the Akaike Information Criterion (AIC). Details of the likelihood function and inferred parameters can be found in Sect 3 of S1 Text.

## Predicting population dynamics from single-cell measurements

Age structured population growth models link the net proliferation rate of the population to single-cell division and death time distributions [38–41]. They relate the exponential proliferation rate of the population to the age dependent division and death rates (hazards) via the Euler-Lotka equation:

$$1 = 2 \int_0^\infty \phi(x) e^{-\gamma x} dx, \tag{1}$$

where $\phi(x) = b(x) e^{-\int_0^x h_{net}(y) dy}$, $h_{net}(y) = b(y) + \mu(y)$ is the net hazard, and $b(y)$ and $\mu(y)$ are the age dependent birth and death hazards respectively. $\gamma$ is the unique root to Eq 1 that gives the population growth rate. The hazard function is a time-dependent generalization of the rate parameter in exponential distributions, given by the ratio of the probability density and survival function (see Sect 8 in S1 Text for details). Before drug administration, the distribution of intermitotic times (IMT) of the cells was fit to an Exponentially Modified Gaussian (EMG) using Maximum Likelihood Estimation (details in Sect 8 in S1 Text). This provided the IMT probability density function, which in turn allows calculation of the survival function and hazard at any given instant in time. The same analysis was performed post drug on IMT and AT (apoptosis time) distributions to get the hazards for division and death respectively. The total hazard (for any event) post drug was then simply calculated as the sum of division and death hazards. The hazards were then used to solve the integral Eq 1 to obtain the population growth/decay rate $\gamma$. Detailed descriptions of parameter inference and calculation of the hazard and proliferation rates are provided in Sect 8 in S1 Text.

## Stochastic simulations of age-structured population dynamics – Model *M0*

Model *M0* corresponds to the conventional age-structured model of population dynamics which generalizes constant division and death rates to age-dependent rates, giving rise to exponential growth at long times. Stochastic simulations of this model were performed for both pre-drug (only division) and post-drug (division and death) scenarios (details can be found in [18,39]. In brief, single-cell distributions of division and death times were taken from the experimental datasets and converted to age-dependent birth and death rates or hazards. For pre-drug simulations, each new cell in a lineage was assigned an age 0 at birth and for each following time-step, the probability of dividing was computed based on the age-dependent division hazard at that time. For post-drug cells with the additional fate of cell death, the total age-dependent hazard at any time was given by the sum of division and death hazards. These hazards were used at each time-step to compute the probabilities of dividing or dying for each extant cell independently. To compute the probabilities, the hazards were used as follows: the probability of either event (division or death) happening in a time step $\Delta t$ was calculated using the net survival probability $exp(-(h(t; \Theta_m^{aft}) + h(t; \Theta_a^{aft}))\Delta t)$, where $h(t; \Theta_m^{aft})$, $h(t; \Theta_a^{aft})$ denote the post-cisplatin division and death hazards of a cell respectively, and $\Theta_m^{aft}$, $\Theta_a^{aft}$ denote the parameter vectors parameterizing the post-drug IMT and AT distributions respectively. Once an event was destined to occur, the choice between division or death for that cell was generated based on the ratio $h(t; \Theta_m^{aft})/(h(t; \Theta_m^{aft}) + h(t; \Theta_a^{aft}))$. For cellular events before addition of the drug, only the hazard function $h(t; \Theta_m^{bef})$ was used, where $\Theta_m^{bef}$ denotes the parameter vector of the pre-drug IMT distribution. The time step $\Delta t$ was chosen such that it was much smaller than the average times of the IMT and time to death distributions. At each division event, two new cells were stored as nodes in the extant cells list and the graph that was uniquely defined for each lineage tree. Attributes like 'birth time', 'state', 'fate', etc.

were stored in these newly created nodes for future reference. Further details can be found in Sects 8 and 9 in S1 Text.

## Competing risks analysis of exponentially distributed division and death time distributions

Since cells exposed to cisplatin can undergo either division or death (apoptosis), these events are mutually exclusive and hence give rise to 'competing risks' for each cell. A well known problem in statistics [42], the competing risks effect leads to significant biases in the experimentally measured times to division and death after cisplatin addition [18]. Since only the event with a shorter time will ultimately be observed for any single-cell, the measured division and apoptosis times will be biased towards lower values, thereby hiding the actual underlying distributions of division and apoptosis times. These underlying distributions therefore are not directly measurable, but need to be inferred from the measured (but biased) division and apoptosis times. A simple analytical calculation using exponential waiting times for the birth and apoptosis events illustrates the competing risks problem, and demonstrates how inference of the underlying distributions can be carried out (see Sect 7 in S1 Text for details). If the underlying independent probability densities of times to mitosis and apoptosis are given by $f_m(t) = \lambda_m e^{-\lambda_m t}$ and $f_a(t) = \lambda_a e^{-\lambda_a t}$ respectively, then the corresponding measured distributions will have faster decaying exponentials, exhibiting the bias towards shorter times (Sect 7 in S1 Text). From a single cell microscopy experiment measuring these biased waiting times $t_i$ and the number of mitosis and death events ($N_m$ and $N_a$ respectively), it can be shown that accurate estimators of $\lambda_m$ and $\lambda_a$ – parameters of the unbiased distributions, are given by:
$\lambda_m^{\inf} = \frac{N_m}{\sum_{i=1}^{N_m + N_a} t_i}$ and $\lambda_a^{\inf} = \frac{N_a}{\sum_{i=1}^{N_m + N_a} t_i}$ (see Sect 7 in S1 Text for details).

## A Markov Chain Monte Carlo approach to competing risks analysis for HCT116 and U2OS cells.

While exponential distributions allow for an analytical solution of the competing risks inference problem, non-exponential distributions along with the pre and post-drug scenarios in the HCT116/U2OS experiments complicate the inference problem and preclude analytic solutions. For the post-drug scenario we needed to find the underlying distributions of IMT and AT as mentioned in the competing risks section, which are modeled as EMGs having three parameters each ($\mu$, $\sigma$ and $\lambda$). We performed the inference of these distributions from the biased IMT and AT values obtained from the experiment using a Metropolis-Hastings Markov Chain Monte Carlo approach. In this method we sampled a total of 8 parameters from their posteriors. Six of these parameters describe the two distributions, IMT and AT ($\mu$, $\sigma$ and $\lambda$ for each distribution). The remaining 2 parameters describe the probability of the cell to enter a quiescent stage and the delay time for the drug to act. To find the posteriors of these parameters, we formulated likelihoods for all the division and death events that are possible post-drug and assumed a uniform prior to obtain the posterior using Metropolis Hastings MCMC. Details of the inference method are provided in Sect 8 in S1 Text.

## Beyond age-structured populations: Stochastic simulations of Models *M1*, *M2* and *M3*

Model *M1*: In Model *M1*, we defined the cells to exist in one of two states before drug administration, one being susceptible to die (sensitive, *S'*) while the other persists (pre-DTP, *P'*) in the face of drug treatment. The cells were allowed to interconvert between these states in drug naive conditions. *S'* cells at time of cisplatin addition were assigned the death fate with 100%

probability. Cells in the *P'* state at time of cisplatin were assigned fates to divide or survive (without dividing), with some probability for each fate. The times taken for the division or death events were sampled from the respective distributions measured in the experiments. This model could recapitulate the experimentally observed lineage correlations and biphasic exponential decay, but not the drug-concentration dependence of the decay rates.

Model *M2*: In this model, the cells did not possess any 'states' before cisplatin administration. As a result, at time of drug addition, every cell was statistically identical, and received randomly assigned fates (die, divide or survive without dividing) with probabilities that depended on the drug concentration. Higher probability of death was assigned for higher drug concentrations. While this model could explain the population decay rates observed in the experiment as well as biphasic decay, it failed in explaining the correlations observed in the final post-cisplatin fates of lineage related cells.

Model *M3*: This model is essentially a combination of Models *M1* and *M2*, where we simulated cells in two states *S'* (sensitive) and *P'* (pre-DTP) prior to drug addition with inter-state transitions. After the drug, the cells were assigned fates from 3 possibilities – death, survival or division, with probabilities conditioned on their state at the time of drug treatment. The state-dependent fate probabilities were chosen from a fate-matrix *F*. If the fate was division or death, the waiting time to the fate was generated by using either the experimentally measured post cisplatin division or death hazard respectively, using the probabilistic framework described above for simulating Model *M0*. Note that unlike in *M0*, here only one appropriate hazard is used post-cisplatin because of the fate assignment using *F*, and no competition between events occur. If a cell was assigned division as the fate, the two resultant daughters maintained the original state of the mother cell at time of drug addition, and the fate assignment process was repeated using the same matrix *F*. The entries of *F* that gave the closest match of simulations with experiments were: for state *S'* – $P(\text{die}|S') = 0.9125$, $P(\text{div}|S') = 0.05$ and $P(\text{stay}|S') = 0.0375$ while for state *P'* – $P(\text{die}|P') = 0.35$, $P(\text{div}|P') = 0.1$ and $P(\text{stay}|P') = 0.55$. This model could quantitatively recapitulate both the lineage correlations as well as the drug concentration dependent decay rates measured in the experiments. Further details are provided in Sects 9 and 10 in S1 Text.

## Calculation of Barcode abundance and diversity from Model *M3*

Model *M3* was used in the barcoding simulations with parameters given above. To mimic barcodes, we simply provide lineage numbers to the initial ancestor cells that the simulations of Model *M3* are started with. These lineage numbers are then maintained across cell divisions, mimicking the inheritance of barcodes. To calculate the barcode abundance, the frequencies of each of the barcodes were extracted just before cisplatin addition as well as 72 hours post cisplatin exposure.

Let $n_i$ be the number of times the $i^{th}$ barcode is observed. Then the probability $p_i$ of observing the barcode is given as

$$p_i = \frac{n_i}{\sum_i n_i} \tag{2}$$

The Shannon Diversity Index (SDI), which gives a quantitative measure of barcode diversity, was then calculated as

$$\text{SDI} = -\sum_i p_i \log(p_i) \tag{3}$$

The difference in the SDI before and after drug then provides the change in diversity of barcodes. This procedure was repeated 100 times to generate error bars in the change in SDI. Further details are provided in Sect 11 in S1 Text.

### Luria-Delbrück analysis of HCT116 dataset and Model *M3*

For both the HCT116 dataset and simulated lineages from Model *M3*, extant cells at time of drug administration were identified. These cells were then followed to determine their end fate at the last time frame of the experiment (survival versus death). If a cell divided post drug, then randomly one daughter was chosen. As a result, the number of cells whose fates were identified equaled the number of cells extant at the time of drug addition. From these cells, lineages were identified and number of persisters per lineage was determined to compute the index of dispersion (variance by mean). This procedure was repeated on the same dataset (originating either from the HCT116 experiment of Model *M3*) 100 times to generate error bars on the index of dispersion.

## Results

### Subpopulations with distinct proliferation rates cannot be detected in HCT116 or U2OS cells before drug

To obtain accurate single-cell intermitotic times (IMTs), apoptosis times (ATs) and cell lineages, we obtained datasets from two previous cell-line studies on colon cancer HCT116 [18] (Fig 1a–1c and Table 1) and osteosarcoma U2OS [37] (Fig 1d–1f and Table 2). These studies performed five days of single-cell time lapse microscopy, tracking cells in two days of drug-free medium followed by three days of cisplatin treatment, with snapshots taken every 30 mins. While HCT116 were exposed to a single concentration of cisplatin approximately near its IC50 value [18], U2OS cells were treated with three concentrations of cisplatin – 7, 10 and 13 $\mu$M corresponding to Low, Medium and High labels [37]. The HCT116 dataset had both the daughter cells of dividing cells tracked (Fig 1a) allowing extraction of single-cell times and fates for all the cells in the population as well as lineage relationships between the single cells. The U2OS dataset however was generated by tracking a single randomly chosen daughter after each cell division (Fig 1d). Besides measurements of IMT and AT of single cells, these datasets allow simultaneous extraction of the population dynamics by counting the number of surviving cells in each time frame (Fig 1b and 1e). Detailed statistics for these two cell types can be found in Tables 1 and 2. In both datasets there were a small fraction of cells which never divided through the entire period they were tracked ('never-dividers'; see Tables 1 and 2). Among these never-dividers in the HCT116 dataset, a higher fraction (10/13) died post cisplatin while the rest survived, reflecting the higher fraction of cell death in the overall population (Table 1). In the U2OS dataset, 2/9, 8/13 and 3/9 never-dividers died post low, medium and high cisplatin respectively (Table 2). Overall, since these never-dividers did not show any clear bias towards any particular end fate (survival or death post cisplatin), and their numbers were low, we excluded these cells from all further analysis. We also excluded a very small fraction of cells that die before cisplatin treatment in the U2OS dataset (Table 2).

We first asked whether distinct sub-populations of slowly dividing cells could be detected in the drug naive HCT116 and U2OS cells (after elimination of the never-dividers). To find the population-time curve of the HCT116 cells (Fig 1b), we summed the number of cells in each time frame. The semi-log plot of the curve shows characteristic exponential growth before the addition of the drug (vertical line in Fig 1b) and a bi-phasic exponential decay after cisplatin exposure. The first larger decay rate switches to a plateau after the 100th hour of the experiment, driven by the drug tolerant persisters in the population. We identified persisters as those cells surviving till the end of the experiment (resulting in the second phase of exponential population decay) and extracted the IMTs of their drug-naive ancestors by tracking back in time along their lineages (Fig 1a, blue line). We compared these drug-naive ancestor

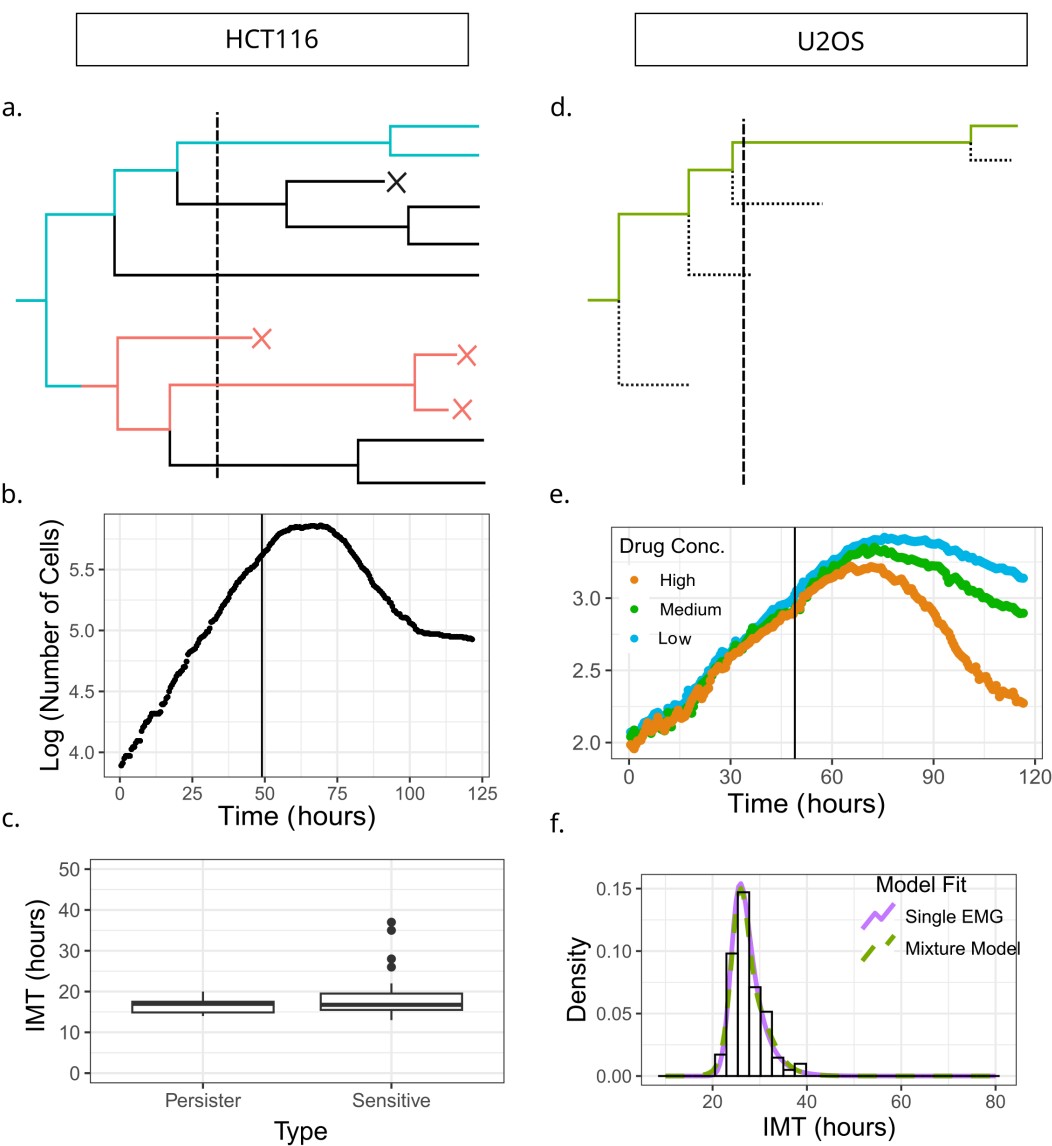

**Fig 1. Pre-cisplatin heterogeneity in inter-mitotic times arises from a single population of cycling cells.** (a) Schematic of a single cell lineage from the HCT116 experiment, where both pairs of sisters were tracked after cell division. The dashed vertical line shows time of administration of cisplatin. Blue lines track back to ancestors of surviving (persister) cells while the red lines track back to ancestors of cells that die (sensitive cells). (b) log of population versus time for HCT116 cells shows a distinct bi-phasic exponential decay after cisplatin (denoted by the vertical line). (c) Boxplot comparing IMTs of distinct drug naive ancestor cells belonging to lineages of persister or sensitive phenotypes. (d) Schematic of the U2OS data where tracking records only one randomly chosen daughter cell after a division event (forward lineages). (e) log of population versus time for U2OS cells with three different cisplatin concentrations. The biphasic decay observed in panel (b) is not observed here. (f) Pre-cisplatin single-cell IMT distribution is better explained with a single EMG (Exponentially Modified Gaussian) model than a mixture of 2 EMGs as verified by comparing AIC values for the two models. The histogram represents the IMT distribution obtained from the single-cell microscopy experiment while the lines are the EMG fits.

IMTs to those of the sensitive cells that died after cisplatin treatment (Fig 1a, red line), and could not detect any statistically significant difference between these two distributions (Fig 1c; see Sect 3 in S1 Text for details).

**Table 1. A summary of the HCT116 dataset.** Detailed reasons for exclusion of the 18 lineages are provided in Table A in S1 Text. 13 out of these 18 are cells that are not observed to divide over the full course of their observed trajectories. 65 single cell lineages proliferate to generate 275 cells at time of cisplatin administration. Out of these 275, 123 die while 52 divide with both daughters ending up dying, resulting in about 64% cell death.

| HCT116 dataset | |
|---|---|
| Total lineages in the dataset | 83 |
| Excluded lineages | 18 |
| Included lineages | 65 |
| Included lineages present at first time point | 49 |
| Included lineages that appear later in time | 16 |
| No. of cells that died pre-cisplatin | 0 |
| No. of cells that did not divide pre-cisplatin (never-dividers) | 13 |
| Never dividers that died post cisplatin | 10 |
| Never dividers that survived post cisplatin | 3 |
| Number of cells present at time of cisplatin addition ($N_{cisp}$) | 275 |
| Number of cells (among $N_{cisp}$) that divided post cisplatin | 112 |
| Number of cells (among $N_{cisp}$) that died post cisplatin | 123 |
| Number of cells (among $N_{cisp}$) that survived post cisplatin | 40 |
| Number of cells (among $N_{cisp}$) that divided twice post cisplatin | 0 |
| Number of daughter pairs formed post cisplatin that both survived | 35 |
| Number of daughter pairs formed post cisplatin that one survived and one died | 25 |
| Number of daughter pairs formed post cisplatin that both died | 52 |

**Table 2. A summary of the U2OS dataset.** The reason for excluding a small fraction of the cells was because they either died pre cisplatin administration, or they never divided.

| U2OS dataset | Low | Medium | High |
|---|---|---|---|
| Cells at T = 0 | 247 | 262 | 316 |
| Excluded cells | 15 | 22 | 20 |
| Included cells | 232 | 240 | 296 |
| No. of cells that died pre-cisplatin | 6 | 9 | 11 |
| No. of cells that did not divide pre-cisplatin (never-dividers) | 9 | 13 | 9 |
| Never dividers that died post cisplatin | 2 | 8 | 3 |
| Never dividers that survived post cisplatin | 7 | 5 | 6 |
| No. of tracked cells at time of cisplatin addition ($N_{cisp}$) | 232 | 240 | 296 |
| Number of cells (among $N_{cisp}$) that divided post cisplatin | 100 | 75 | 67 |
| Number of cells (among $N_{cisp}$) that died post cisplatin | 30 | 66 | 176 |
| Number of cells (among $N_{cisp}$) that survived post cisplatin | 102 | 99 | 53 |
| Number of cells (among $N_{cisp}$) that divided twice post cisplatin | 2 | 2 | 1 |

For the U2OS population-time curves, calculated by averaging over the number of cells across 51 fields of view per time frame, showed a characteristic exponential growth before cisplatin addition and concentration-dependent exponential decay post cisplatin exposure (Fig 1e). However, five days of tracking was not sufficient to observe the second slower exponential decay unlike in the HCT116 cells. While this lack of a second slower exponential decay may appear to be limiting for further analyses, this turns out not to be the case. As explained in the subsequent sections, predicting the first exponential decay is the most important mathematical task in this work, which allows developing new models of cell fate. Further, to investigate whether multiple sub-populations with distinct IMT distributions exist pre-cisplatin, we fit the pre-drug IMT distribution with a mixture model (Sect 3 in S1 Text) instead of tracing the ancestors of persisters, thereby circumventing the need for a second exponential phase. We thus compared the ability of single and two-component Exponentially Modified Gaussian (EMG) models to explain the IMT distribution of drug naive cells,

using the Akaike Information Criterion (AIC). We found that the single EMG (AIC = 3687.4) fared better than the mixture model (AIC = 4137.649). Also, the two models generated best fit distributions that were essentially indistinguishable, suggesting the absence of multiple sub-populations with distinct proliferation rates within the U2OS population as well (Fig 1f).

Our analysis demonstrates that in both HCT116 and U2OS cell types, after setting aside the never-dividers, the heterogeneity in cell cycle times before addition of cisplatin is best explained by a single population of cycling cells, not by multiple sub-populations with distinctly different cycling kinetics. This suggests that if a pre-DTP state exists amongst these cells before drug administration, it does not have any fitness cost as compared to the sensitive cell state.

## Proliferation rate of cancer cells before drug can be predicted from single-cell division times

Having established that a single IMT distribution best explains the datasets before cisplatin treatment, we next asked how accurately this IMT distribution can predict the population growth rates. For exponentially proliferating cells, the population growth rate is approximately given by the inverse of the mean of the IMT distribution. While this is the most popular model used to describe exponential cellular growth, the underlying assumption here is that the IMT distribution is exponential. However, this is clearly not true in our datasets (Fig 1f) and hence a more general approach is required. An age-structured population model (Model *M0*; see Methods and Sect 9 in S1 Text for details) is a generalization of the simple exponential growth models, that allows for non-exponential IMT distributions, and connects the population growth rate to the full IMT distribution via the Euler-Lotka equation (see Methods and Sect 4 in S1 Text for more details) [38,39,41,43]. From the measured single-cell IMTs we estimated the population growth rates from both the inverse of the mean as well as the Euler-Lotka equation. We compared these predictions with the true growth rate, defined as the linear regression slope of the semi log plot of number of cells as a function of time (Table 3). In both HCT116 and U2OS datasets, there were only about 3% or less cells that died before cisplatin treatment (Tables 1 and 2), hence only the IMT distributions were used in predicting the population growth rates before drug.

For the HCT116 cells, we found that the inverse of the mean gave an error of around 62% (percentage difference between estimated versus true growth rates), while the Euler-Lotka estimate gave a much reduced error of 13% (Table 3 and Fig 2a). The large error in the former model was not surprising as the measured IMT distribution was neither exponential nor very narrow to be effectively represented by the mean. For the latter result, we further investigated the potential origins of the 13% error. To check whether this error could arise simply due to the small number of cells in our experiments, we performed stochastic simulations taking the measured IMT distribution as input to generate *in silico* proliferation trajectories with similar cell numbers.

For each simulation run we calculated the growth rate prediction errors using both the growth rate estimates, mimicking the procedure we carried out on the experimental data. Since the age distribution of the cells at the initial time is an input to the simulations, and it is known that this distribution can affect the emergent population dynamics [39], we estimated the age distribution of the initial cells directly from the experimental data (Fig 2b). Based on the IMT distribution of the full pre-cisplatin population, we inferred the starting ages of the initially plated cells in the experiment using a Maximum Likelihood approach (Fig 2b; details in Sect 6 in S1 Text). We then repeated the simulations using the inferred initial age distribution and measured IMT distributions and found that the Euler-Lotka errors

**Table 3. An age structured model (Euler-Lotka equation) accurately predicts population growth rates of both HCT116 and U2OS cells before cisplatin treatment.** Shown here are deviations (in %) of growth rates predicted by either the Euler-Lotka equation or the inverse of the mean IMT, from the experimentally measured growth rate (obtained via linear regression of the cell-population versus time curve in Fig 2a and 2d). The growth rate prediction by the Euler-Lotka equation for HCT116 cells is within ~11% of the error expected from age-structured population simulations with the inferred starting age-distributions. For the U2OS cells, the predictions are off by about ~ 20%.

| Model for pre-drug growth rate prediction | HCT116 | U2OS |
|---|---|---|
| Inverse of mean IMT | 62% | 73% |
| Euler-Lotka | 13% | 23% |
| Expected Errors (Simulations) | $\sim 2\%$ | $\sim 4\%$ |

peaked around 2% (Fig 2c). We carried out a similar analysis for the U2OS dataset, where we once again found a large error of around 73% from the inverse of the mean prediction, and a much reduced error of 23% from the Euler-Lotka equation (Fig 2d and Table 3). We found that the inferred initial age distribution of the starting U2OS cells was broader than that of the HCT116 cells (Fig 2e), and when used in the simulations, could account for about 4% error in the growth rate prediction (Fig 2f). The unexplained errors of about 10 – 20% for HCT116 and U2OS cells could be due to artefacts of the image analysis pipeline, or from other unaccounted biological sources such as cell-size control of the cell cycle which might induce lineage correlations in cell cycle times and consequently modifications to the growth rate [44].

Our results in this section demonstrate that in the absence of cisplatin, the age-structured model predictions of population growth rate using a single IMT distribution is accurate to within 10 – 20% in both HCT116 and U2OS cell types. This provides a benchmark for the expected error level when trying to predict population dynamics from our single-cell datasets. Additionally, the fact that a single IMT distribution is sufficient to predict the population growth rate is consistent with results from the previous section, which suggested the absence of sub-populations of cells with different cycling kinetics.

## First exponential decay of drug treated cancer cells is not established by single-cell division and death times

Since the age-structured model (Model *M0*) could predict pre-cisplatin population growth rates from the measured cell division times within 20% error, we next asked if a similar prediction accuracy can be achieved in the post-cisplatin scenario, where an added dimension is cell death causing the populations to decay instead of grow. Throughout this section, we will be focusing only on the first phase of exponential decay after cisplatin treatment (for the U2OS cells the experiments were not long enough to observe the second exponential phase). To find the IMT and AT distributions for both HCT116 and U2OS cells (Fig 3a–3d), we extracted the times from cells that were either born before drug exposure but divided or died after drug exposure, or cells that were both born and reached a fate after drug exposure. The primary reason for having to include cells born before drug administration was because there were very few cells born after cisplatin exposure that went on to finish a full cell cycle to divide again. There were no HCT116 cells that finished a full division event purely post cisplatin (Table 1), while in the U2OS datasets there were just 2/232, 2/240 and 1/296 such cells in the Low, Medium and High concentrations respectively (Table 2). A careful deconvolution of the pre versus post drug cell cycle times from these 'straddling cells' therefore needs to be performed via inference [18], as we describe later in this section.

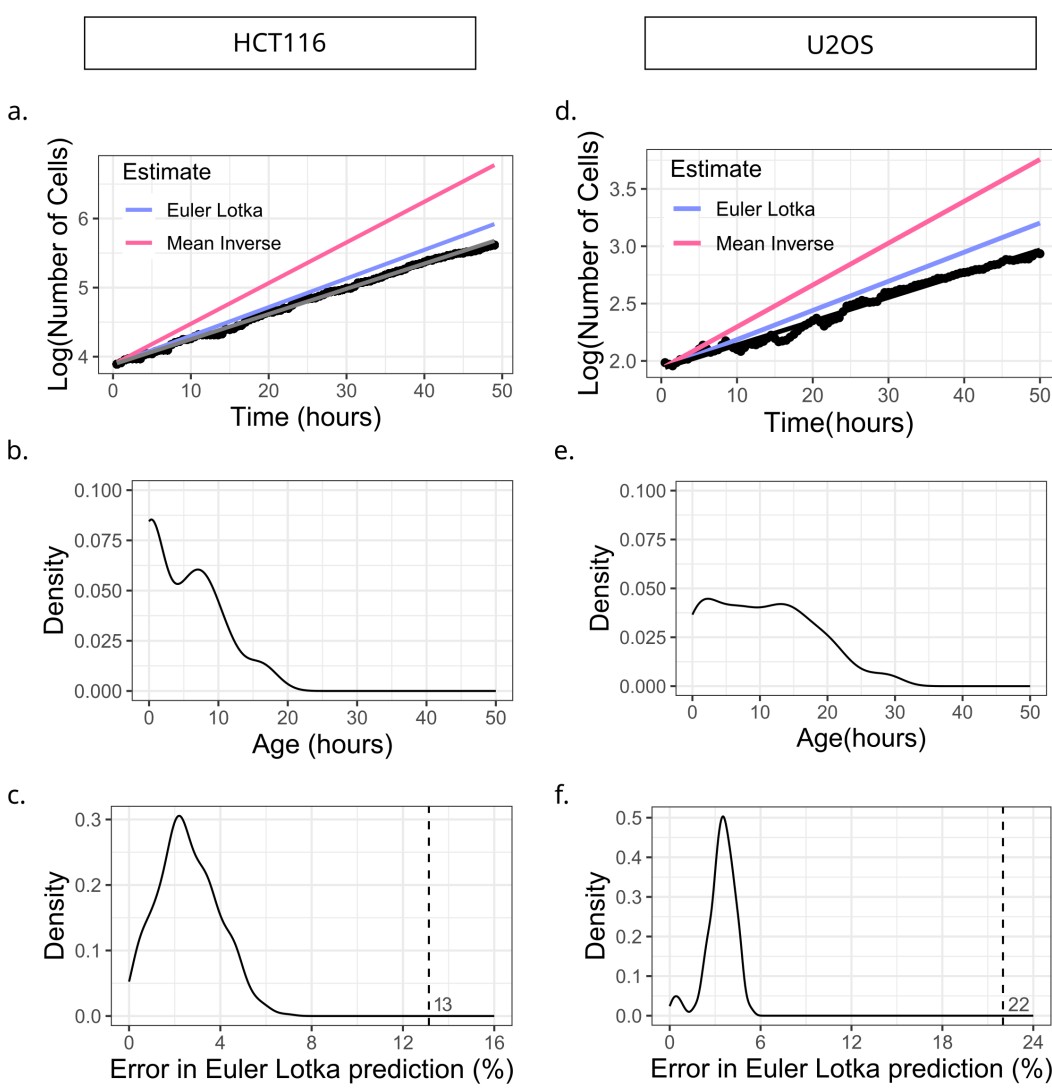

**Fig 2. An age-structured model accurately predicts population growth rates from single-cell IMT distributions.** (a)-(c) are plots for HCT116, (d)-(f) are for U2OS. (a) Semi-log plot of population versus time curve of HCT116 cells before cisplatin administration. Predicted growth rate using inverse of the mean (pink line; error = 62%) and the Euler-Lotka equation (blue line; error = 13%) are compared with the experimental data shown as black dots. (b) Maximum Likelihood inference of the age distribution of ancestor cells immediately after seeding. (c) Distribution of errors in Euler-Lotka predictions of growth rate, obtained by running 500 iterations of an age-structured population simulation with initial age distribution taken from panel (b). Vertical dashed line represents the Euler-Lotka prediction error in experimental data (corresponding to error of the blue line in panel (a)). (d) Semi-log plot of population versus time curve of U2OS cells before cisplatin, with colours as in panel (a). Predicted growth rate using inverse of the mean (pink line; error = 73%) and the Euler-Lotka equation (blue line; error = 22%) are compared to the experimental data shown in black dots. (e) Inferred age distribution of seeded population of U2OS cells, as in panel (b). (f) Distribution of errors in Euler-Lotka estimates, obtained by running 500 iterations of simulations as in panel (c). Initial age distribution was taken from panel (e) for these simulations. Vertical dashed line represents Euler-Lotka prediction error in experimental data (corresponding to error of the blue line in panel (d)).

Though the U2OS decay rates were distinctly different between High ($-0.0249 \pm 0.0004\,h^{-1}$), Medium ($-0.0119 \pm 0.0002\,h^{-1}$) and Low ($-0.0088 \pm 0.0001\,h^{-1}$) cisplatin, which amounts to a 3-fold change between High and Low concentrations, surprisingly there were no statistically

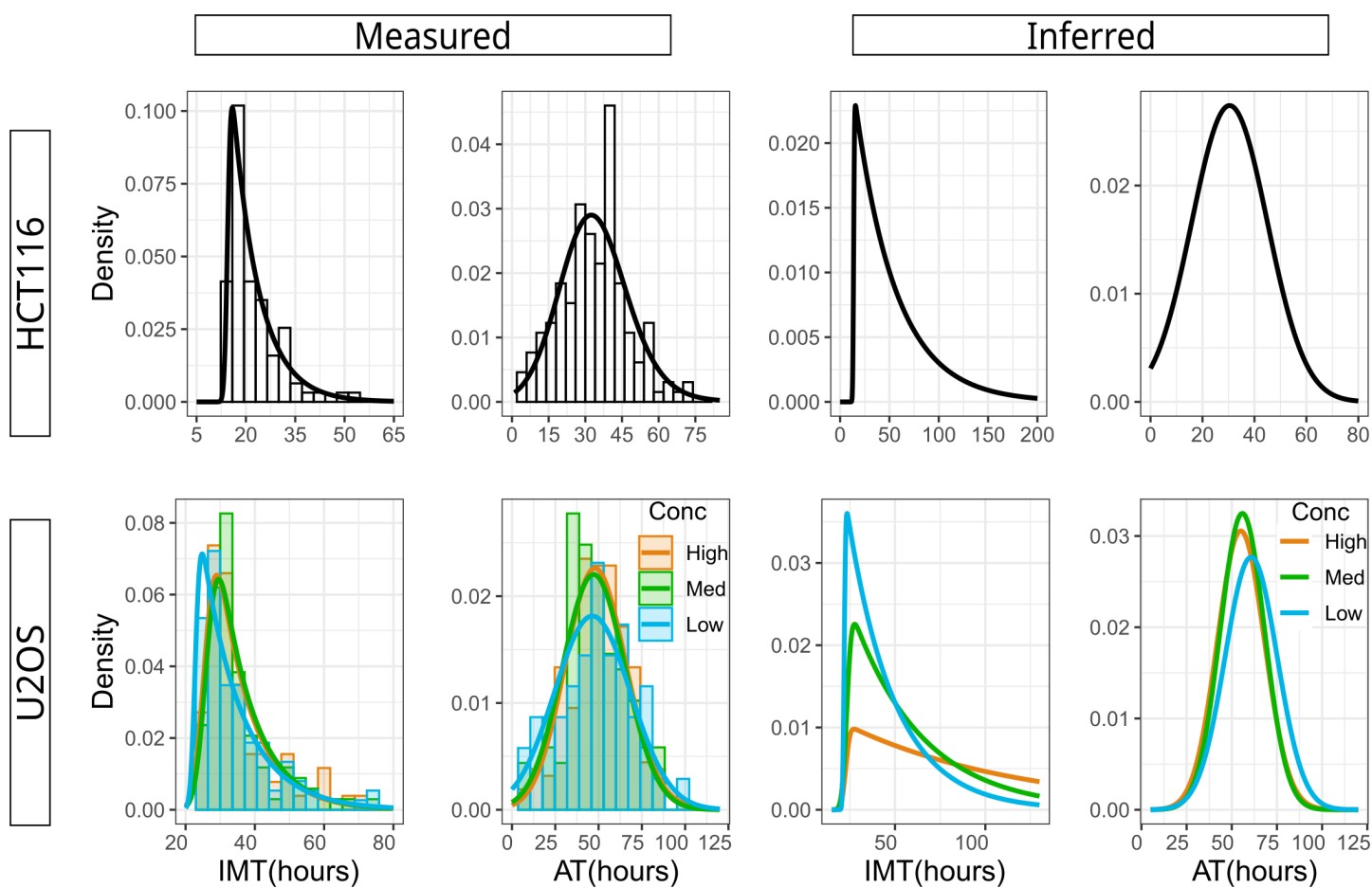

**Fig 3. Inferring the underlying post-drug IMT and AT distributions from the experimentally measured, biased distributions.** (a)-(d) are plots of measured IMT and AT distributions of HCT116 and U2OS cells while (e)-(h) are the competing-risks corrected distributions inferred using a Markov Chain Monte Carlo approach (see Methods and Sect 8 in S1 Text for details). (a)-(b) IMT and AT distributions of HCT116 cells. All cells that divided or died during the treatment period were tracked and measured, including cells that were born before and after cisplatin addition. (c)-(d) Forward lineage distributions for U2OS cells that have division and death events that occurred during the treatment period. The three distributions in both panels (c) and (d) are almost identical, demonstrating that increasing drug concentration does not affect division or death rates post cisplatin. For the three IMT distributions, Kruskal-Wallis test gave a p-value of 0.22 while for the three AT distributions, Kruskal Wallis test gave a p-value of 0.53. (e) Inferred IMT distribution of HCT116 cells. Compared to panel (a), the long tail is evident demonstrating the significant bias in the measured distribution. (f) Inferred AT distribution of HCT116 cells, which is similar to the measured distribution in panel (b). (g) Inferred IMT distributions for U2OS cells reveal increasingly long tails for increasing cisplatin concentrations. (h) Similar to panel (f), the inferred AT distributions for U2OS cells is almost identical to the measured AT distributions.

significant differences in the measured IMT and AT distributions across the three cisplatin concentrations (Fig 3c and 3d respectively). This directly showed, in a model-independent manner, that times to division or death post cisplatin (and hence the division and death rates) do not determine the population decay rates. This result contradicts the widely used assumption that the effect of drugs can be well described either as increasing the cellular death rate or reducing the division rate [31–36].

To rigorously and quantitatively confirm that the minor differences in the measured IMT and AT distributions cannot explain the large differences in decay rates across cisplatin concentrations, we calculated the predicted rates using both inverse of the mean and Model *M0* (see Sect 9 in S1 Text for details). The IMT and AT distributions were not sufficient to accurately predict the concentration-dependent decay rates of the U2OS cells, producing errors

greater than 100% ('Measured' columns in Table 4). Indeed, even for HCT116 cells where we had data from one drug concentration, the error in predicted population decay rates was greater than 100% ('Measured' column in Table 4). Taken together, these findings were rather surprising in light of our previous section results as well as the vast body of literature modeling population growth/decay as the difference in birth and death rates.

We therefore wondered if our analysis using the measured IMT and AT distributions suffered from a competing-risks bias, a statistical effect that prevents direct measurement of the correct underlying distributions when the events are mutually exclusive. We have previously demonstrated that this effect leads to strong biases in the IMT and AT distributions measured after drug treatment [18], and hence can potentially lead to erroneous population growth rate predictions. A simple analytic calculation for exponentially distributed IMT and AT is presented in Sect 8 in S1 Text that provides intuition for this competing risks effect, and how it can be corrected using inference techniques. In the more complex case of non-exponentially distributed IMT and AT as is the case with the HCT116 and U2OS datasets, we used a Markov Chain Monte Carlo (MCMC) approach to infer the parameters of the unobserved distributions (see Sect 8 in S1 Text for details). The inferred IMT distributions of HCT116 cells (Fig 3e) showed a longer right tail (slower division times) compared to the measured IMT distribution, which was not observed for the AT distribution (Fig 3f). For the U2OS cells, the right tails of the inferred IMT distributions increased with higher drug concentrations (Fig 3g) while once again the inferred AT distributions (Fig 3h) had similar right tails compared to the measured AT. Using these inferred IMT and AT distributions we then recalculated the predicted decay rates using the mean inverse and the Eula-Lotka equation. While the errors significantly reduced ('Inferred' columns in Table 4), they were still very high with the smallest error around 40% in both HCT116 and U2OS and the largest error greater than 100% for low and medium cisplatin doses in U2OS.

In summary, we found that estimates of population decay rate using measured single-cell times to division and death post cisplatin treatment were erroneous by over 100%, predicting positive growth instead of decay. Even after correcting for competing risk biases, the population decay rates could be predicted at best with an error of 40%, often with errors as high as 100%. These counter-intuitive results suggest that measured single-cell IMT and AT are not sufficient to estimate population decay rates post addition of cisplatin. In turn these results suggest that *post*-drug stochastic competition of cell fates inherent to these exponential growth models (Model *M0*), is likely to be an incorrect model of cell-fate decisions in the development of cancer drug persistence.

**Table 4**. **Age structured models fail to predict population decay rates for cells treated with cisplatin.** Shown here are deviations (in %) of the predicted decay rates from the measured decay rates (obtained via linear regression of post-cisplatin experimental data in Fig 1b and 1e). Predictions were made using IMT and AT distributions obtained directly from the experiment ('Measured') and from competing risks-corrected unbiased distributions ('Inferred').

| | HCT116 | | U2OS | | | | | |
|---|---|---|---|---|---|---|---|---|
| | Measured | Inferred | Measured | | | Inferred | | |
| **Model** | | | Low | Med | High | Low | Med | High |
| Mean Inverse | > 100% | 51.8% | > 100% | > 100% | > 100% | > 100% | 85% | 60.2% |
| Euler-Lotka | > 100% | 40.3% | > 100% | > 100% | > 100% | > 100% | > 100% | 40.3% |

## Cell-state inheritance with fate decisions before *and* during drug treatment control cancer population dynamics

We next asked if the nature and timing of fate decisions can be determined by developing a series of models with varying assumptions and checking for compatibility with experimental data. The large errors in Model *M0*'s predictions of decay rates suggest that the cell fates are likely decided either *during* or *before* drug addition, largely independent of the single-cell IMT and AT distributions. Therefore we developed three alternate models of the fate decision process, Models *M1*, *M2* and *M3* (Fig 4). In these models, cell fates are either (i) determined based on 2 cell-states before drug – a pre-DTP state *P'* that results in surviving persisters post drug (some of which can divide), and a sensitive state *S'* that leads to cell death after drug administration (Model *M1*), (ii) decided randomly for each cell at the time of drug addition with probabilities that depend on the drug concentration (Model *M2*), or (iii) a combination of both *M1* and *M2* where pre-existing cell-states determine the end fate with large probability while the drug concentration modulates this probability to some extent (Model *M3*). *M1* therefore describes a purely pre-drug fate decision scenario while *M2* describes purely drug-induced fate decisions, effectively modeling a scenario where persister cells emerge exclusively after drug treatment, in a concentration dependent manner.

*M3* combines elements of both models *M1* and *M2*, and captures the existence of cell states before drug that predispose cells towards survival or persistence post drug. Inter-conversion between states S' and P' in Models *M1* and *M3* is modeled using a Markov Chain described by a transition matrix *T* while the drug concentration-dependent fate probabilities in Model *M3* are imposed using a fate matrix *F* (Fig 4). In effect, *F* models cell-state switching post drug treatment, by allowing a small fraction of pre-drug S' cells to persist and P' cells to die post drug treatment. Further details and parameterizations of each model are provided in Methods and Sects 9 and 10 in S1 Text. We found that Models *M1* and *M2* could recapitulate only certain aspects of the experimental datasets (see S1 for details). *M1* generated clear biphasic exponential decays and could explain the lineage correlations, but not the concentration-dependence of the first exponential decay. This is because every cell is already fated to either die, or survive/divide based on its cell state at time of drug addition. The drug itself plays no role in determining fate. *M2* on the other hand could generate biphasic exponential decays that were drug concentration dependent, but could not generate lineage correlations due to the lack of cell-states. Only Model *M3* could quantitatively explain all three major observations simultaneously: (1) decay rate for a fixed drug concentration, (2) lineage correlations in end-fate (Fig 5a) and (3) drug concentration dependence of the population decay rates. On simulating *M3*, where the pre-drug transitions create a steady-state distribution of S' and P' cells (Fig 5b), we noticed that a broad spectrum of effective decay rates could be obtained after cisplatin exposure even with the IMT and AT distributions fixed. This arises either due to different Fate matrices for a given Transition matrix (Fig 5c) or due to different Transition matrices for a given Fate matrix (Fig 5d). This explains the surprising results from Fig 3c–3d where the IMT and AT distributions were found to be insensitive to drug concentration – the IMT and AT distributions have little contribution towards establishing the decay rate. Rather, the population dynamics is dominated by the relative proportion of the two states at time of drug exposure (determined by *T*), and the entries of *F*. Finally, by performing a coarse parameter sweep we found that for certain physically realistic values of the *T* and *F* matrices, we could approximately recapitulate the experimentally measured decay rates (Fig 5e) as well as the lineage correlations (Fig 5f) while maintaining the same measured IMT and AT distributions as observed in the HCT116 experiment.

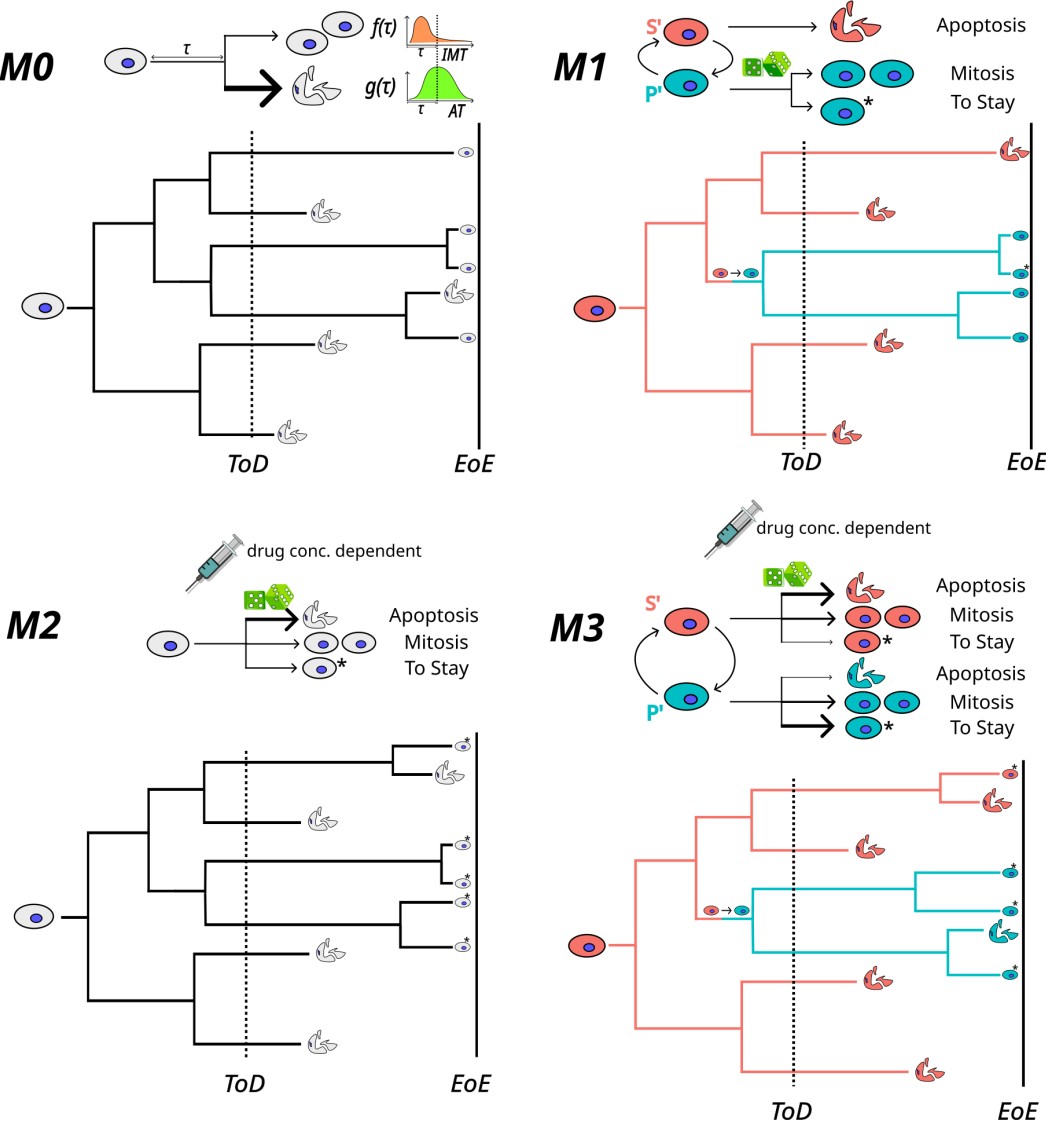

**Fig 4. Schematics of models *M0* to *M3*, highlighting the differences in timing and mode of cell fate determination of each model.** The models are shown with the possible state and state-to-fate transitions on top, along with an example lineage below. Cells in gray indicate no role of cell-states in determination of fate. Coloured cells indicate pre-existing cell-states that pre-determine final fates to a large extent. The dice indicates a probabilistic state-to-fate map, that assigns fates stochastically to each single cell, conditioned on its state at the time of drug administration. A syringe accompanying the dice indicates drug concentration dependent state-to-fate probabilities. The state-to-fate map is modeled with a 'Fate Matrix' *F*. Thicker arrows indicate higher probabilities in the *F* matrix. 'ToD' is Time of Drug and 'EoE' is End of Experiment. In *M0*, fate decisions happen entirely post-drug via hazards of division and death drawn from the post-drug IMT and AT distributions. In models *M1-M3* the IMT and AT distributions are essentially irrelevant, except for setting an overall timescale. In *M1*, fate decisions are based purely on pre-existing cell states. In *M2*, single cells get assigned their fate at time of drug administration based on drug concentration-dependent probabilities. *M3* is a combination of *M1* and *M2*, where fates are largely pre-determined by cell states, but get modulated at time of drug administration to some degree. For example, as depicted in the lineage in Model *M3*, cells in state *S'* at ToD mostly die post drug, but a few survive.

The fate probabilities for cells to die, divide and stay alive for the two states were: Sensitive state *S'* = (0.9125, 0.05, 0.0375); pre-DTP state *P'* = (0.35, 0.1, 0.55). These numbers support our expectation that though cells in the sensitive state are more likely to die (with

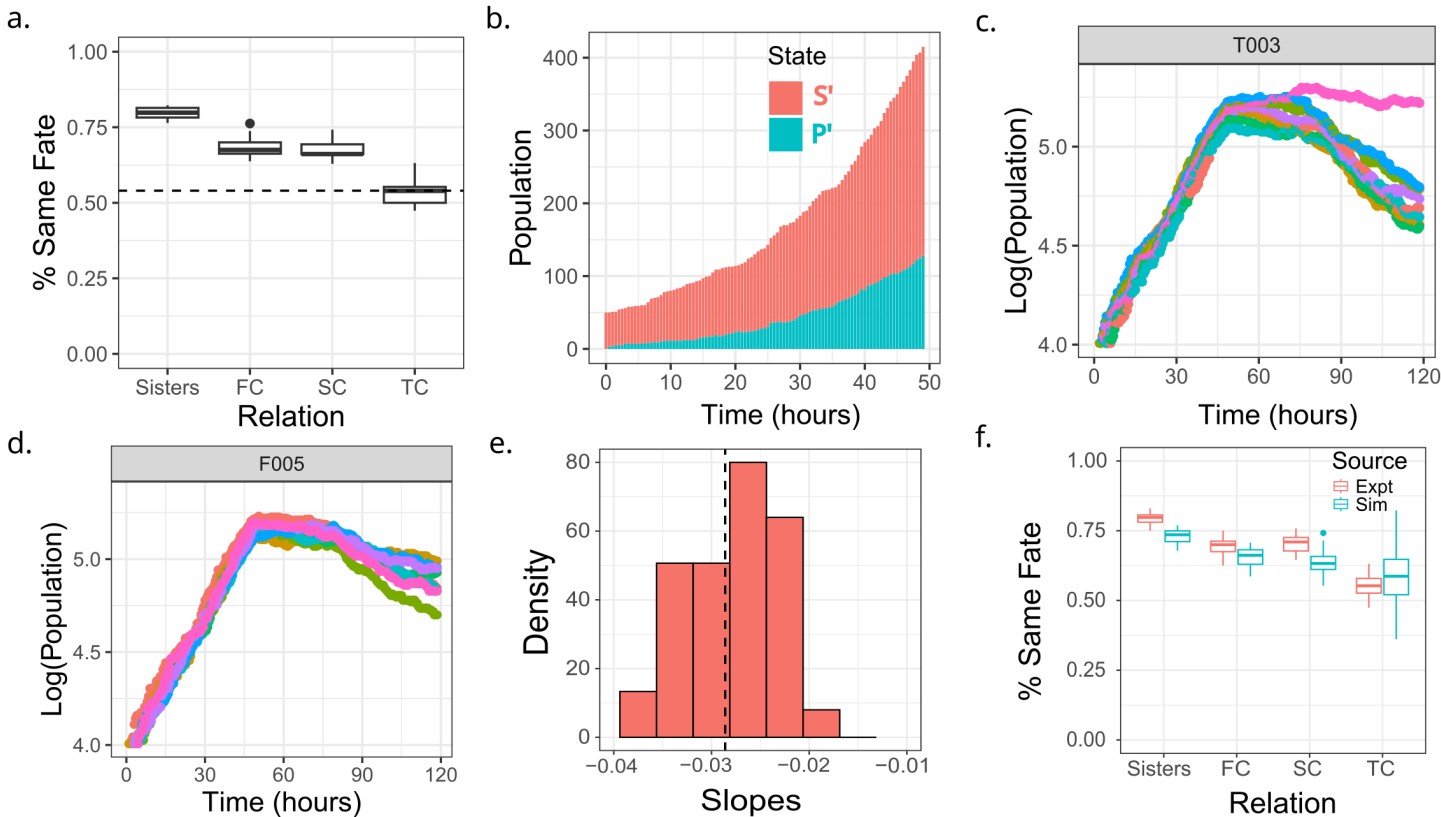

**Fig 5. A two state Markov model with state-dependent fates (Model *M3*) quantitatively explains the experimentally measured decay rate and correlations in lineage related HCT116 cells.** (a) Lineage correlations in fates of HCT116 cells measured from time lapse experiments. Correlations exist for first and second cousins (FC and SC), while third cousins (TC) are similar to randomly selected pairs of non-lineage related cells (horizontal dashed line). (b) An example run of Model *M3* showing the relative populations of the two states as a function of time before drug administration. (c) Population versus time trajectories from simulations of Model *M3* using identical IMT and AT distributions, a fixed Transition matrix *T*, but different Fate matrices. Each colour corresponds to simulations with a different *F*. (d) Same as panel (c), but with different Transition matrices for a fixed Fate matrix *F*. (e) Distribution of decay rates from the simulations (red histogram). The vertical dashed line is the decay rate obtained from the HCT116 experimental dataset, $-0.0286\,\mathrm{hr}^{-1}$. (f) Comparison of lineage correlations obtained from the HCT116 experiment (red) with simulations of Model *M3* (blue). Transition rates used in the simulation were symmetric for transition between the two states, $9 \times 10^{-3}$ per hour. The fate probabilities for cells to die, divide and stay alive for the two states were: Sensitive state *S'* = (0.9125, 0.05, 0.0375); pre-DTP state *P'* = (0.35, 0.1, 0.55).

probability 0.91) and pre-DTP state more likely to survive (probability 0.55), with some small but finite probability *S'* cells survive (probability 0.037) and *P'* cells die (probability 0.35). Note however that these numbers should be treated only as approximate estimates, with further work required to develop accurate parameter inference from these datasets.

Our analysis in this section quantitatively explains why the birth and death rates (and more generally the single-cell IMT, AT distributions) do not determine the decay rates of drug treated cells at the population level. We demonstrate that the transition rates between cell states before drug treatment, the ability of these states to be passed down across cellular generations, and the cisplatin concentration-dependent probabilities of these states to result in death or survival largely determine the population dynamics and lineage correlations of cells after drug treatment.

## Change in barcode diversity before and after drug cannot establish timing of fate decisions in the presence of cell-state switching

Our analyses thus far demonstrated the importance of pre-existing cellular states in the population that leads to persistence of cells in the face of cisplatin treatment. This conclusion was true for both colorectal cancer as well as osteosarcoma cell types. However, a recent study on colorectal cancer cells concluded that the persister fate arose after the addition of cisplatin, based upon analysis of a high-complexity barcode library that had been transduced into the cancer cells [25]. In short, this study found that the barcode diversity (measured by the Shannon Diversity Index) showed no reduction upon cisplatin treatment, which suggested that the persister fate choice was likely made post drug treatment. Since these results are conceptually different from our findings, we next investigated potential reasons that may explain the apparently contradictory results.

The barcoding approach to determine the timing of persister fate choice relies on comparing the barcode diversity, or abundance of unique barcodes, before and after drug exposure in many cellular replicates. A reduction in barcode diversity after drug treatment is interpreted as selection of a small set of pre-existing persister cells (Fig 5a i). Maintenance of barcode diversity on the other hand suggests a scenario where persister cells arise randomly in cells after drug treatment, thereby leading to many surviving barcodes (Fig 5a ii). However, these interpretations were originally derived in the context of genetic mutations [45] where reversal of mutations is rare. The same assumptions do not hold in case of non-genetic cell states (the pre-DTP state $P'$ or sensitive state $S'$ in our case), and we argued in the previous sections that transitions between the two states is a fundamental aspect of the dynamics in both HCT116 and U2OS cell types.

To quantitatively explore how cell-state switching before drug addition could affect interpretations of barcode diversity, we used Model *M3* with the same parameters from the previous section, with the additional information of barcodes (Fig a iii–iv; see Sects 11 and 12 in S1 Text for details). In brief, each starting cell was given a barcode ID that was inherited by all descendants across cell divisions and maintained over cell-state changes. The fates were conditioned on cell-state as described before for Model *M3*. The results of the simulation are shown as an abundance bubble plot (Fig b) for qualitative visualisation and as a Shannon Diversity Index (SDI) plot (Fig c) for quantitative assessment of barcode enrichment or diversity. The drop in Shannon Diversity Index post drug was observed only when the simulation did not allow for state transitions before drug treatment, which is equivalent to very slow transitions compared to a typical cell cycle time (Fig b–c, 'No Trans'). However, if the transitions were allowed with rates that gave rise to the experimentally measured lineage correlations (Fig 5f), it resulted in no change in the SDI value (Fig b–c, 'SimExpt'). This short analysis confirmed that the timing of development of pre-DTP states cannot be inferred using current barcode analysis techniques and might require development of novel methods to account for cell-state transitions.

Finally, we used an orthogonal approach to analyse the timing of fate decisions for the HCT116 cells – a Luria-Delbrück (LD) fluctuation test. For U2OS cells this analysis was not possible since the second exponential decay phase was not observed, precluding identification of persisters (Fig 1e). The LD test is based on quantifying the Variance over Mean Ratio (VMR) of the persister counts across lineages post cisplatin. A scenario where the persister fate is assigned with probability $p$ at time of drug administration, and the persisters do not proliferate post drug, would lead to a Binomial distribution in the number of persisters and a consequent VMR given by $1-p$. On the other hand, if the persister fates were pre-determined based on pre-drug inheritable cell-states, then the VMR would be greater

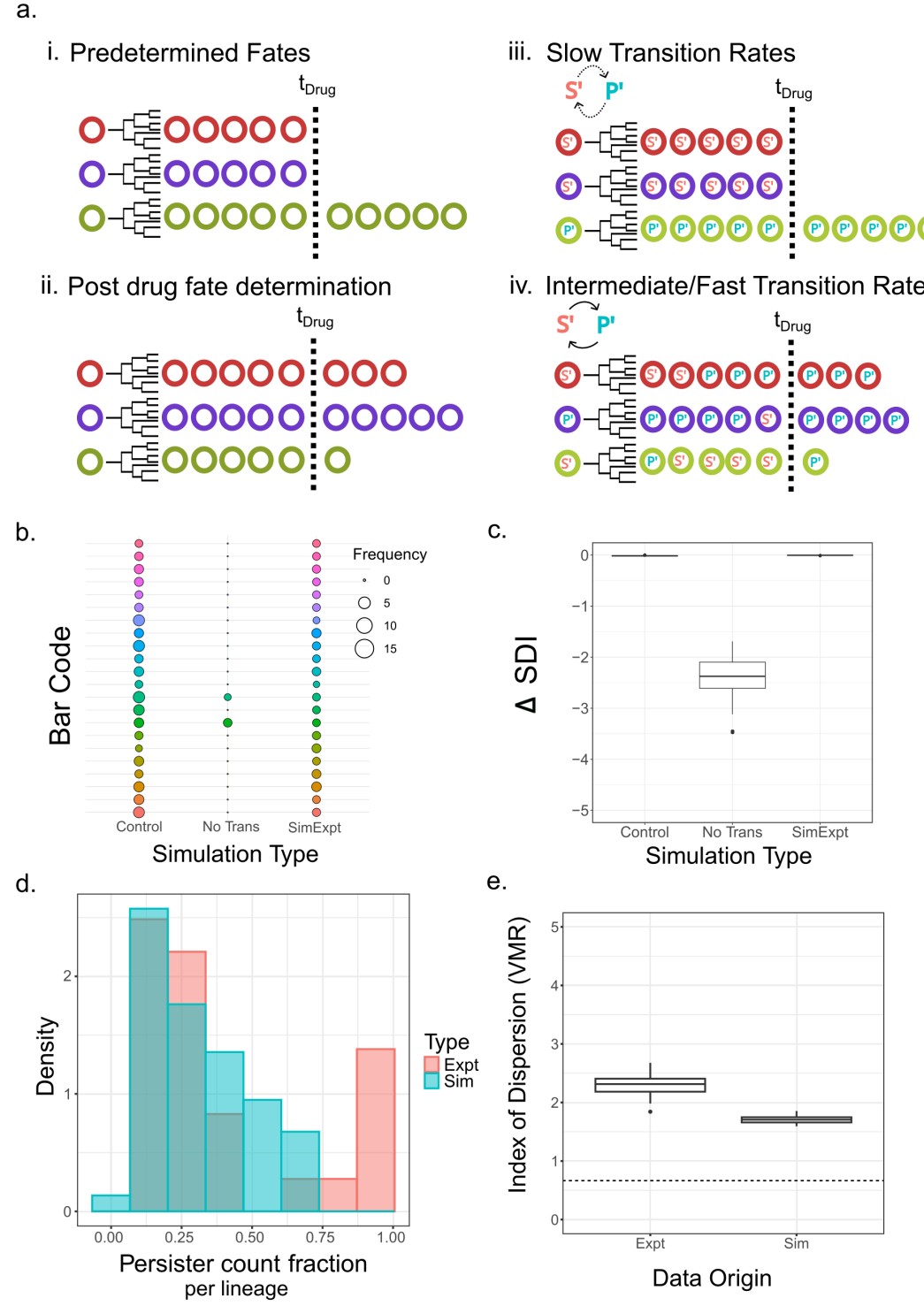

**Fig 6. Maintenance of barcode diversity before and after drug cannot be interpreted as a signature of drug-induced cell fates.** (a) Current barcoding methods relate fate determination to pre-drug cell-states if the barcode diversity, shown here by the presence of distinct barcodes (colored circles), significantly reduces after drug (i). However, if the diversity does not reduce then fate determination is assumed to be in response to drug exposure (ii). Our model however, suggests that both the conditions (i & ii) can be obtained with fates largely determined before drug administration. Simulation results show that in conditions when the cell-states before drug transition on time scales much slower than the cell division time, barcode diversity shrinks significantly post drug treatment (iii). However, if the transition rates are of the order of a few cell cycles or faster, more sensitive cells transition to a pre-DTP state, resulting in insignificant change in barcode diversity

post-drug (iv). (b) Barcode abundance plot post drug treatment obtained from simulations of Model $M3$. Circle fill colors represent distinct barcodes while their sizes show barcode frequency counts. Results of 3 simulations are compared. 'Control': simulations with cell-state independent fate assignment of cells; 'No Trans': simulations with cell-fate assignment conditional on cell-states that do not inter-convert; 'SimExpt': simulations with state dependent fate assignment with cell-state transitions allowed before drug addition. (c) Diversity of barcodes can be measured by the Shannon Diversity Index. The boxplot here shows change in Shannon Diversity Index for the corresponding simulations described in panel (b). (d) Distribution of persister count fractions across lineages in the HCT116 dataset and Model $M3$ simulations. (e) Variance over Mean Ratio (VMR) of persister cell counts as measured in the experiment (Expt) and the Model $M3$ simulation matching experimental results (Sim). The horizontal dashed line shows the VMR for a Binomial distribution with $p = 0.36$

than 1 [46]. We therefore calculated the VMR from both the HCT116 experimental dataset as well as our simulations of Model $M3$. To ensure that cell divisions post drug did not bias our calculation of the VMR, we identified all extant cells at the time of drug administration and tracked their final post-drug fates, randomly choosing only one of the two daughters in cases where the cell divided post drug (see Methods and Sect 13 in S1 Text for more details). Using this method, we first checked that the Model $M3$ generated similar fractions of persister cells across lineages as in the HCT116 data (Fig d). Next, we computed the VMRs: for the HCT116 data, the Binomial distribution predicts a VMR of 0.64, since the fraction of persisters was approximately $p = 0.36$ (see Table 1; out of 275 cells at time of cisplatin, 123 cells died while 52 divided and then died). However, the calculated VMR was around 2.4 (Fig e; see Sect 13 in S1 Text for details). Model $M3$ simulations with parameters that reproduced experimentally observed decay rates (Fig 5e) and lineage correlations (Fig 5f), generated a VMR of 1.6 (Fig e). Therefore both the experimental data and simulations suggest that end fates could not have been assigned solely as a post-drug phenomenon, consistent with the results of our analysis using lineage correlations. In summary, the results in this section show that change in barcode diversity before and after drug treatment is not always sufficient to establish the timing of fate decisions. This is true when fates are related to non-genetic cell-states which inter-convert between each other on time scales similar to the cell-cycle time. Maintenance of barcode diversity after drug treatment is therefore consistent with a pre-DTP state existing before drug treatment and driving fate outcomes, as our analyses demonstrates using both lineage correlations as well as the Luria-Delbrück fluctuation test.

## Discussion

In this work we demonstrate how general principles underlying the dynamics of drug persistence can be discovered in the process of connecting measured single-cell kinetics to the emergent population dynamics of cancer cells. Quite unexpectedly, the single-cell measurements directly show that increasing drug concentration does not affect the birth or death rates (Fig 3c–3d), unlike the widespread assumption in the field [31,33–36]. These results suggest that stochastic competition between fates based on drug-induced cellular birth and death rates is not the mode of decision-making underlying the emergence of non-genetic drug tolerance. Rather, we demonstrate that non-genetic cell-states existing before drug treatment largely bias the eventual fates towards drug-tolerance or susceptibility. These biases are further modulated at the time of addition of the drug, leading to a quantitative explanation of both the population dynamics as well as the lineage correlations observed in experimental datasets of HCT116 and U2OS cells (Fig 4f–4g). While earlier studies have discussed the role of pre-existent cell-state switching in cancer drug tolerance, they typically use markers of specific cell-states to demonstrate state-switching and emergence of drug tolerant persisters

[3,47–50]. Our work however is based on more general arguments for the existence of non-genetic states and transitions between them, requiring no *a priori* knowledge of the molecular architecture of the underlying states. This is particularly powerful since the molecular details of these states remain unknown in all generality, and are likely to comprise transcriptional, post-transcriptional as well as epigenetic elements. Additionally, to the best of our knowledge this work is the first to quantitatively explain the connection between measured single-cell kinetics and the emergent population dynamics of drug-treated cancer cells.

In the widely used exponential growth models and their generalizations, the fraction of cells that die (establishing the first decay rate) depend entirely upon the parameters of the post-drug IMT and AT distributions. However, in the presence of non-genetic cellular states that pre-determine fate outcomes (sensitive and pre-DTP states), the fraction of cells that die no longer depends exclusively on the IMT and AT distributions, but also on the transition rates of these pre-existing cell states. Though this idea has been suggested in many previous theoretical works that typically analyze FACS datasets [3,36,51–55], it has not been directly demonstrated from single-cell measurements linking IMT and AT distributions to the population dynamics. Additionally, the idea that single-cell division and death kinetics may not be sufficient to predict drug-treated population dynamics has been suggested before in the context of lung cancer cells [56]. However, this study did not incorporate distributions of time to death or correct for competing-risks biases, thereby making it challenging to interpret the results.

Furthermore, the implications of cell-fate lineage correlations have been poorly appreciated in the context of non-genetic heterogeneity and the timing of emergence of pre-DTP states. While early studies on TRAIL-induced apoptosis had already suggested that lineage correlations imply early fate decisions[2,17], a careful study later explored the detailed implications of correlations in apoptosis and their decay across sisters, cousins and higher order lineage-related cells [16]. Based on careful time-lapse microscopy experiments, this study separated out the effects of spatial correlations potentially arising from inter-cellular signalling from correlations arising from inheritance of cell-states, to argue for decisions of apoptosis taken well before addition of the drug [16]. Recently we demonstrated that correlations also exist in the dynamics of signaling pathways that get upregulated only after drug treatment (rate of p53 increase in response to cisplatin), suggesting that even if cellular states change upon drug addition, the propensity for these changes are largely pre-programmed into upstream regulatory networks of the ancestor cells existing before drug addition [18]. Taken together with our current study, the emerging evidence points towards a scenario where cell-states existing well before drug addition largely determine eventual cell fates after addition of drugs. Interestingly, recent tour-de-force studies using clever combinations of high-complexity barcoding and single-molecule FISH have arrived at broadly similar conclusions. These newer studies have also begun to shed light on important and previously unavailable information on the genes whose expression states pre-determine the ultimate fate of cancer cells in response to drugs[19,21], and further demonstrate that transcriptional states post long-term drug treatment may be very different from those that existed prior to drug exposure [29,57].

An intriguing implication of our results is that both the pre-DTP as well as the drug-sensitive states exist in cells that are actively proliferating. This suggests that DTPs need not arise exclusively from quiescent (also called dormant) or slowly cycling cells (such as cancer stem cells), which is primarily the evidence that has been reported in the literature from bacteria, mammalian cell lines, primary cells as well as tissues [12,27,58–62]. Importantly, our inability to find multiple proliferative sub-populations in either HCT116 or U2OS cells

before drug treatment (Fig 1c and 1f), combined with the fact that we could accurately predict the population growth rates from the measured single IMT distribution (Fig 2c and 2f), demonstrates that the pre-DTP state does not have any fitness cost. This result is intriguing since a fitness cost is typically associated with drug resistance arising from genetic mutations [35,63,64], and has recently been suggested for non-genetic persister states as well [28]. It is interesting to note however that though the IMT distributions do not distinguish the ancestors of persisters versus sensitive cells, the number of divisions undergone can be predictive of eventual cell fate after drug treatment [37].

The use of single-cell time lapse microscopy datasets in this study provides an advantage over other barcoding studies in terms of the high resolution at which lineages can be reconstructed. Indeed, these results demonstrate the advantage of using single-cell resolved lineage correlations over barcode-diversity based methods to infer timing of fate decisions (Fig 5b–5c). However, these experiments and hence our accompanying analyses have the major limitation of exploring very short time-scales after drug treatment (on the order of a few days). As a result, we cannot comment on effects of long-term drug exposure on cell-state switching, or drug induced gradual cell-state reprogramming over long time scales [5–7,20,29]. Careful combinations of time-lapse microscopy along with barcoding studies will be required to delineate short-term versus long term dynamics and provide a full picture of non-genetic drug tolerance that eventually leads to genetic resistance.

## Conclusions

We quantitatively demonstrate in this study that stochastic choices between survival and death based on post-drug division and death rates is *not* how cancer cells make fate decisions. Rather, the choices are largely pre-determined via cell-states that exist at least 2-3 generations before drug treatment, and are inherited by successive generations. This is true both for the cell-state that likely leads to drug-tolerant persisters as well as the state that makes cells susceptible, thereby suggesting that the notion of a fitness cost need not necessarily apply to non-genetic mechanisms driving persistence. To a smaller extent the fate decisions are also modulated by the drug, and together with the pre-existing cell-states, quantitatively explain the emergence of lineage correlations in end-fate and the population dynamics. We argue that the existence of lineage correlations is a better alternative to inferring the timing of fate decisions as opposed to barcode diversity analysis, which is hard to interpret in the presence of cell-state switching. These arguments do not require knowledge of the specific molecular details of the underlying cell-states, and hence provide a general and powerful approach to the study of drug-tolerance on short time scales driven by non-genetic heterogeneity.

## Supporting information

**S1 Text. Supporting information text, figures and tables combined.**
(PDF)

## Author contributions

**Conceptualization:** Shaon Chakrabarti.

**Data curation:** Anton Iyer, Adrián E. Granada, Shaon Chakrabarti.

**Formal analysis:** Anton Iyer, Adrian Alva, Shaon Chakrabarti.

**Funding acquisition:** Shaon Chakrabarti.

**Investigation:** Anton Iyer, Adrian Alva, Shaon Chakrabarti.

**Methodology:** Anton Iyer.

**Project administration:** Shaon Chakrabarti.

**Resources:** Adrián E. Granada.

**Software:** Anton Iyer, Adrian Alva.

**Supervision:** Shaon Chakrabarti.

**Writing – original draft:** Anton Iyer, Shaon Chakrabarti.

**Writing – review & editing:** Anton Iyer, Adrián E. Granada, Shaon Chakrabarti.

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
