## [Decision Letter · Decision Letter 0]

18 Mar 2025

PCOMPBIOL-D-25-00043

Inheritable cell-states shape drug-persister correlations and population dynamics in cancer cells

PLOS Computational Biology

Dear Dr. Chakrabarti,

Thank you for submitting your manuscript to PLOS Computational Biology. After careful consideration, we feel that it has merit but does not fully meet PLOS Computational Biology's publication criteria as it currently stands. Therefore, we invite you to submit a revised version of the manuscript that addresses the points raised during the review process.

Please submit your revised manuscript within 60 days May 18 2025 11:59PM. If you will need more time than this to complete your revisions, please reply to this message or contact the journal office at ploscompbiol@plos.org. Please include the following items when submitting your revised manuscript:

We look forward to receiving your revised manuscript.

Kind regards,

John P Barton

Guest Editor

PLOS Computational Biology

Mark Alber

Section Editor

PLOS Computational Biology

**Additional Editor Comments :**

The reviewers have undertaken a careful reading of the paper, raising important technical and conceptual questions. In particular, it is especially important to clearly establish the appropriateness of the data sets and analyses used in this work. The manuscript would also be improved by including a more thorough description of relevant statistics of the data and technical details of the methods.

**Journal Requirements:**

At this stage, the following Authors/Authors require contributions: Anton Iyer, Adrian E Granada, and Shaon Chakrabarti. Please ensure that the full contributions of each author are acknowledged in the "Add/Edit/Remove Authors" section of our submission form.

5) We have noticed that you have uploaded Supporting Information files, but you have not included a list of legends. Please add a full list of legends for your Supporting Information files after the references list.

Potential Copyright Issues:

i) Figure 4A. Please confirm whether you drew the images / clip-art within the figure panels by hand. If you did not draw the images, please provide (a) a link to the source of the images or icons and their license / terms of use; or (b) written permission from the copyright holder to publish the images or icons under our CC BY 4.0 license. Alternatively, you may replace the images with open source alternatives. See these open source resources you may use to replace images / clip-art:

7) Thank you for stating “All code will be available on https://github.com/Shaonlab.” We strongly recommend all authors decide on a data sharing plan before acceptance, as the process can be lengthy and hold up publication timelines. Please note that, though access restrictions are acceptable now, your entire data will need to be made freely accessible if your manuscript is accepted for publication. This policy applies to all data except where public deposition would breach compliance with the protocol approved by your research ethics board. If you are unable to adhere to our open data policy, please kindly revise your statement to explain your reasoning and we will seek the editor's input on an exemption. Please be assured that, once you have provided your new statement, the assessment of your exemption will not hold up the peer review process. 

8) Please amend your detailed Financial Disclosure statement. This is published with the article. It must therefore be completed in full sentences and contain the exact wording you wish to be published.

9) Your current Financial Disclosure states, "National Centre for Biological Sciences core funds".

However, your funding information on the submission form indicates receiving no funds. Please ensure that the funders and grant numbers match between the Financial Disclosure field and the Funding Information tab in your submission form. Note that the funders must be provided in the same order in both places as well.

**Reviewers' comments:**

Reviewer's Responses to Questions

Reviewer #1: In their manuscript "Inheritable cell-states shape drug-persister correlations and population dynamics in cancer cells", Iyer at all study the important problem of drug tolerant persister cells, which is germane to both cancer and infectious disease. Using rigorous modeling/inference and carefully done in vitro experiments, they support their conclusions well, and even challenge the model with data from single cell barcoding experiments. The writing is clear, the results support the conclusions, and the work provides an important springboard for future development/discovery. Honestly, I don't have any complaints. Well done.

Reviewer #2: In this very interesting and extensive manuscript, the authors utilize single-cell tracking from two cancer cell lines to investigate distribution of intermitotic times during cisplatin treatment. The authors link a theoretical framework for cellular growth with the single-cell data and population dynamics during treatment. Amongst a number of interesting conclusions, the authors conclude that the time to division or death is not influenced by cisplatin treatment. The authors also use their theoretical framework to investigate recent results in colorectal cancer cell lines that used barcoding to study the distribution of drug-tolerant persistors in response to chemotherapy. The manuscript reads as if it was written for a primarily biological audience and (possibly), a different journal (as there is no author summary, the bibliography is in a non-PLoS format and a few other formatting differences, including the bibliography). I find the manuscript suffers for the formatting decision to move all the technical and theoretical methods to the Supplemental Information and I struggled to understand the results using only the methods in the Main Text. Consequently, I suggest that the authors consider moving some of their theoretical work and model formulations to the main text.

I have a number of questions regarding the current manuscript.

How were the probabilities of death or division events calculated and transformed into hazard rates for the age structured model?

How is the delay time for the drug estimated? As the authors are considering the intermitotic and apotosis time distributions, I think it is important to explore how these distributions depend upon the assumed drug-effect delay. How is this delay time implemented in the models and when estimating the IMT/AT?

The authors excluded a small population of cells that did not divide or die during the in vitro experiments. This decision, which is only stated briefly at the end of page 7, seems to be a fairly strong imposition on the cell-line data that the authors are modelling. In effect, one of the main results of the manuscript is that a single exponentially perturbed Gaussian distribution is sufficient to characterize the intermitotic times of the cell populations. As the authors have discarded data from a sub-population of cells that are slowly dividing (and thus likely have IMTs following a different distribution), the conclusion that a single IMT is sufficient to describe the data seems to follow directly from this assumption. I would be very interested to see how the results may change if these cells that do not divide or die are not discarded from the analysis (their IMT or AT could be incorporated as censored data.)

The HCT116 cell line is stated as starting with an initial concentration of 65 cells, but figure 1b shows a population of 1 cell; the initial population size is ~0 = log (1). Further, the authors show the single and mixture model for the IMT distribution in Figure 1e without the data. It’s hard to evaluate the fit and I would appreciate seeing the (scaled) raw histograms.

The authors state that using the inverse of the mean of the IMT distribution as a population growth rate does not lead to a good fit with the experimental growth curve and show this in Figure 2. I found this result difficult to parse. The net population growth rate, which is observed as the log(number of cells) involves the net proliferation rate. This net growth rate involves both cell proliferation and cellular loss due to apoptosis. In this respect, it isn’t surprising to me that a net growth rate corresponding to only the mean IMT (and not capturing apoptosis) would over-estimate the true growth curve. Can the authors better situate this result, or more clearly state how they estimated the proliferation rate from the IMT distribution? (Here, I found the manuscript difficult to follow, as the mathematical details are all in the S1 text.)

The authors also repeatedly remark on the “error” between the model and the data. How is this error calculated and defined? Further, they state that using the inferred starting age distribution can “account for 4%” of the error. I don’t know how to understand this statement; how can the authors determine this?

On line 274, the authors note that the decay rates are “distinctly” different between high, low, and control cisplatin treatments. Are these differences statistically significant? Leading to their conclusion that the times to division or death post treatment are not treatment dependent as the IMT and AT distributions only showed “negligible differences”. What does negligible difference mean in this context? Can the authors quantify this statement (both in terms of the decay rates and the “negligible differences” between distributions). Without these differences being cleared defined and reported, it is very difficult to evaluate the claim that these results contradict the long-standing assumption that cytotoxic drugs effect proliferation or apoptosis rates.

On lines 301, the authors remark on a significant “right skew” in the inferred IMT distribution compared to the measured IMT (slower division times.) Is this necessarily the case? I can imagine two distributions with the same mean division time but different 3rd moments. I found it difficult to evaluate this result without a precise mathematical statement.

I was unable to follow the final result section, on barcode labelling in simulations, using the methods in the manuscript. Here, I think the manuscript would benefit from moving much of the technical details currently in the SI to the main text, as it would allow the authors to make precise statements regarding their models that are currently missing.

I took a quick look at the listed Github page and could not find a relevant repository for this manuscript. I appreciate that the authors state that the code will be available on Github, will they also make the underlying data publicly available?

Reviewer #3: Review:

Iyer et al.: ““Inheritable cell-states shape drug-persister correlations and population dynamics in cancer cells””

%%

Summary:

The authors combine time-lapse microscopy data from two cancer cell lines with quantitative modeling to explore the relationship between single-cell kinetics and population dynamics in drug-tolerant persisters. Their key finding is that while single-cell division and death time statistics accurately describe population dynamics before drug exposure, they fail to do so afterward. Surprisingly, they observe that increasing drug concentration leads to a threefold increase in population decay rates, despite no significant changes in single-cell division or death times. To reconcile these observations, the authors propose and validate a model in which cell fate decisions are largely predetermined by pre-existing cellular states inherited across 2–3 generations, rather than being dictated solely by birth and death rates post-drug exposure. They further support this model with orthogonal barcoding data and fluctuation test analysis.

%%

Overall assessment

The manuscript is well-written, engaging, and addresses a highly relevant topic. The authors' data-driven approach is appropriate, insightful, and presents a novel perspective on cancer persister dynamics. Their quantitative analyses appear methodologically sound.

A key strength of this study is its integration of both single-cell and population-level data to inform quantitative models—a relatively novel approach that provides valuable insights into the principles underlying persister cell dynamics.

That said, there are some aspects of the study that would benefit from further clarification and refinement. Indeed, while the authors’ findings have the potential to be highly impactful, I have significant concerns regarding the extent to which the authors' claims are supported by the quality of the data and the pertinence of their analyses and models.

One key point concerns the datasets used. The U2OS dataset, as acknowledged by the authors, does not exhibit biphasic death kinetics—a hallmark of persister cell dynamics. Additionally, partial lineage tracking in this dataset raises further concerns about its appropriateness. Given that this dataset, specifically its concentration-dependent behavior, appears to play a crucial role in the inference and validation analyses, it would be helpful for the authors to explicitly explain why the U2OS dataset remains relevant for understanding persister dynamics despite its apparent a priori limitations.

Furthermore, the lack of clear discussions regarding the data statistics (e.g. number of cells considered in the analyses), and the specific details of the tested models, makes it difficult to fully assess the robustness of the model selection analysis. Comparisons between experimental data, predictions from different models, and simulations with similar statistical constraints are not consistently provided. Additionally, other potential explanations for the authors' findings are not always considered. I believe that a more thorough examination of these aspects would strengthen the authors' claims and provide better support for their conclusions.

Another key point regards the rationale for testing certain models, which is not always evident to me. A key contribution of this study, as I said, is its combined analysis of both single-cell and population data. As such, it is not a priori clear why model M1, for example, should be considered as a viable alternative, given that it assumes persister cells continue growing during treatment and does not seem to reproduce biphasic death kinetics. If some models are a priori clearly incompatible with data, then the model selection analysis—though methodologically sound—becomes somewhat less meaningful. More informative comparisons would involve models that are thought to plausibly capture all key experimental findings.

In this reviewer’s opinion, these concerns represent a significant limitation of the study in its current form. This is particularly important given that some of the study’s conclusions challenge established knowledge in the field. In such cases, it is crucial that claims be supported by robust data and rigorous quantitative analyses. While the manuscript acknowledges some potential limitations, a more explicit discussion of these aspects would improve clarity and enhance the study’s overall impact.

To strengthen the manuscript and ensure its findings are as compelling as possible, I would encourage the authors to undertake substantial revisions addressing these points.

I therefore recommend a major revision of the manuscript.

%%

Specific comments.

1. The manuscript should more effectively and transparently communicate the nature and quality of the data and models considered. While clearly written, it lacks a structured and easily accessible presentation—both textually and visually—of the datasets analyzed. Despite these data originate from the authors' previous work, I strongly recommend adding a dedicated Methods section, along with a corresponding main figure, to explicitly detail the dataset’s key aspects, including the number of cells analyzed, experimental protocols (briefly), and statistical properties.

Since the study’s conclusions heavily rely on these datasets, it is crucial that readers can readily access and interpret this information. Currently, relevant details are dispersed across the Results and Supplementary Information sections, making them less accessible. Beyond the infographics in Figure 1A–D, no other visual representation of the data is provided. A dedicated figure summarizing the dataset could, for instance, incorporate Figure 1A, B, D, and E, supplemented with additional panels outlining the experimental protocol and statistical power. This addition would significantly improve clarity and ensure that the role of the data in shaping the study’s conclusions is immediately evident.

Similarly, Figure 4A, which presents the different models considered, should be extracted and presented as a separate figure dedicated to model descriptions. This would enhance readability and help distinguish between the data, model and results components of the study.

---

2. Can the authors compare model M3 with a more competitive alternative than models M1 and M2? Alternatively, could the authors clarify why M1 and M2 are considered competitive alternatives? For instance, while the authors present their findings as contrasting a more predominant drug-induced scenario, it seems to me that neither model M0 nor models M1 and M2 effectively capture a drug-induced scenario where susceptible cells transition to a persistent state exclusively during treatment and in a drug-dependent manner.

---

3. “Subpopulations with distinct proliferation rates cannot be detected in HCT116 or U2OS cells”  I would specify that this result applies **before treatment**. Same for the next result section “Proliferation rate of cancer cells can be predicted from single-cell division times”

---

4. On line 191 the authors state “we identified persisters as those cells surviving till the of the experiment”. While this is reasonable, can the authors validate this claim by quantifying, for example, the fraction of persister cells predicted by a given model at that point?

---

5. In Figure 1C, the authors conclude that if persisters pre-existed treatment, they should have no fitness cost relative to susceptible cells. While I agree with this reasoning, how can the authors rule out the possibility that, in their data, no persister cells actually pre-exist treatment (i.e., that Figure 1C represents only one cell type=susceptible)? If transitions between the two states can occur between consecutive birth and death events, could it be that a persister cell’s ancestor was actually a susceptible cell? Would it be possible to validate this "ancestor-type reconstruction" analysis through simulations?

---

6. In Figure 2 and Table 1, the authors report that the population growth rate before drug addition can be predicted from the single-cell IMT distribution, albeit with higher estimation errors than expected from simulations of the age-structured model with similar simulated cell numbers. The authors suggest that this discrepancy may arise from artifacts in the image analysis pipeline or unaccounted biological factors, such as size-control effects during cell-cycle progression. Can the authors elaborate more on how size-control might influence their analysis?

Additionally, could the observed discrepancy be partially explained by the omission of cell death in the analysis? It is unclear why cell death was neglected, given that the population growth rate being estimated is the effective growth rate—that is, the net difference between growth and death rates. Or am I wrong? Can the authors clarify the rationale for this choice?

---

7. The finding that single-cell IMT and AT distributions remain unchanged across different drug concentrations is both surprising and intriguing. Can the authors clarify their rationale for including in this analysis the division times of cells born before drug exposure? If the drug primarily affects a specific phase of the cell cycle (e.g., G1 or S/G2/M), some of these cells may not experience the drug's effects and could divide as usual, potentially introducing a bias in the analysis. Would it be possible to replicate the analysis considering only cells born after drug exposure to assess whether this influences the results? Additionally, can the authors rule out the possibility that the drug concentrations used in this analysis are beyond the range where they still affect single-cell division and death rates? As the authors note, it is generally believed that the death rate increases with drug concentration, but at some point, this increase should plateau as the death rate reaches a maximal inducible value. Could the authors test this possibility by comparing model predictions with the observed concentration-dependent behavior at the population level? Notably, a concentration-dependent response at the population level could still emerge from concentration-dependent transition rates (e.g. from susceptible to persister cells).

---

8. Regarding the fluctuation test analysis, could the authors clarify in the main text how the number of persisters was calculated experimentally? Specifically, does it refer to the number of surviving cells in each lineage at the end of the experiment?

Additionally, it seems to me that the pre-existing and drug-induced scenarios do not necessarily imply super-Poissonian and Poissonian statistics for the number of persisters, respectively. The specific statistics followed by the number of persister cells depends on the assumptions made about their dynamics both before and after drug exposure. For example, if persisters neither grow nor die, their number will follow a Poisson distribution, even if they were generated before drug exposure (as their evolution would be a counting process, similar to mutational events in the standard Luria-Delbrück setup). On the other hand, if persisters can grow, their number at a given time after treatment would follow super-Poisson statistics, even if they are generated only in the presence of the drug.

Therefore, it seems important to me to note that the super-poissonian statistics of persisters predicted by model M3 (cf. Fig. 4A) arise specifically because this model assumes persisters can divide during treatment. However, as discussed, this does not necessarily rule out a model where persisters are drug-induced (i.e., generated only after drug exposure) and also can divide during treatment. Can the authors test for this scenario?

Given these considerations, I recommend that the authors provide more precision in describing the models considered in their comparisons and model selection analyses.

**Have the authors made all data and (if applicable) computational code underlying the findings in their manuscript fully available?**

Reviewer #1: Yes

Reviewer #2: **No: **

Reviewer #3: Yes

PLOS authors have the option to publish the peer review history of their article (what does this mean?). If published, this will include your full peer review and any attached files.

Reviewer #1: No

Reviewer #2: No

Reviewer #3: No

**Figure resubmission:**
---

## [Decision Letter · Decision Letter 1]

5 Aug 2025

PCOMPBIOL-D-25-00043R1

Inheritable cell-states shape drug-persister correlations and population dynamics in cancer cells

PLOS Computational Biology

Dear Dr. Chakrabarti,

Thank you for submitting your manuscript to PLOS Computational Biology. After careful consideration, we feel that it has merit but does not fully meet PLOS Computational Biology's publication criteria as it currently stands. Therefore, we invite you to submit a revised version of the manuscript that addresses the points raised during the review process.

Please submit your revised manuscript within 30 days Oct 05 2025 11:59PM. If you will need more time than this to complete your revisions, please reply to this message or contact the journal office at ploscompbiol@plos.org. Please include the following items when submitting your revised manuscript:

We look forward to receiving your revised manuscript.

Kind regards,

John P Barton

Guest Editor

PLOS Computational Biology

Mark Alber

Section Editor

PLOS Computational Biology

**Additional Editor Comments :**

While most queries from the first round of review have been answered, the reviewers have a few remaining questions/points for clarification, as indicated below.

**Journal Requirements:**

1) Please upload all main figures as separate Figure files in .tif or .eps format. For more information about how to convert and format your figure files please see our guidelines: 

2) We have noticed that you have uploaded Supporting Information files, but you have not included a list of legends. Please add a full list of legends for your Supporting Information files after the references list.

3) Please amend your detailed Financial Disclosure statement. This is published with the article. It must therefore be completed in full sentences and contain the exact wording you wish to be published.

3) If any authors received a salary from any of your funders, please state which authors and which funders.

4) Your current Financial Disclosure states, "National Centre for Biological Sciences core funds". However, your funding information on the submission form doesn't indicate receiving any funds. Please ensure that the funders and grant numbers match between the Financial Disclosure field and the Funding Information tab in your submission form. Note that the funders must be provided in the same order in both places as well. 

Please amend the funding information for your study and confirm the order in which funding contributions should appear.

**Reviewers' comments:**

Reviewer's Responses to Questions

Reviewer #2: I appreciate the authors effort in their response and revision, which has addressed many of my earlier concerns. I have a few remaining questions regarding the results and methods in the revised manuscript. As in my first review, I found the methods in the Main Text inadequate to understand the results. As PLOS Comp Biol does not have a page limit and I believe the manuscript will be significantly easier to understand if the mathematical and technical details are presented in the main text, I suggest that the authors consider moving some of their theoretical work and model formulations to the main text.

In the response to my earlier comments, the authors state

"among the 65 lineages of HCT116 cells that we analyzed, there were no cells that died before cisplatin treatment. (with a similar comment regarding the other cell line)"

While I appreciate that this finding strengthens the use of the inverse of the mean of the IMT distribution for comparison against the Euler-Lotka equation, it seems contradictory with the authors conclusion that

"This result contradicts the widely used assumption that the effect of drugs can be well described either as increasing the cellular death rate or reducing the division rate [lines 341 & 342]."

It could be that I'm missing something simple, but I would appreciate clarification on how these two statements can be reconciled (especially as there are certainly cells that die post treatment, as shown in Table 1 & 2).

On lines 264-265, the authors conclude that "best explained by a single population of cycling cells,

265 not by multiple sub-populations with distinctly different cycling kinetics." I think it is worth explicitly restating that this conclusion is drawn by analysing a data-set from which the "never-dividers" were removed. I appreciate the authors response that the "never-dividers" did not show a bias towards death or division. However, the concern in my first review is that, excluding cells that do not divide during the experiment may remove a small subpopulation (~5% of the total population) for which the IMT and AT are not well-described by a single distribution. I appreciate that the behaviour of these cells is both right and left censored and so appropriately accounting for them in the likelihood can be difficult. However, as the main result in this section is that the entire population of cells follows a similar distribution, I think that excluding the 5% of "never-dividers" is not well-justified and should be explicitly stated as a caveat on the quoted conclusion.

Alternatively, I would be very happy to see the authors include these "never-dividers" in their analysis.

In Model M3, on lines 191-192, once the fate decision has been made, is the accumulated life-span of the individual cell considered when sampling the "waiting time" from the appropriate IMT/AT distribution? It seems to me that this is necessarily a conditional probability, as the cell has survived some non-zero time without death/division, which should be accounted for when determining the fate waiting time. Once again, here, I think the manuscript could be improved by including more technical details, particularly given PLOS Comp Biol's readership.

Reviewer #3: I thank the authors for their careful and thorough responses to my previous comments.

Overall, I am satisfied with the revisions made to the manuscript, which adequately address the concerns I raised.

I have no further comments to raise and therefore support the publication of this work in PLOS Computational Biology.

**Have the authors made all data and (if applicable) computational code underlying the findings in their manuscript fully available?**

Reviewer #2: Yes

Reviewer #3: Yes

PLOS authors have the option to publish the peer review history of their article (what does this mean?). If published, this will include your full peer review and any attached files.

Reviewer #2: No

Reviewer #3: No

**Figure resubmission:**
---

## [Editor Report · Decision Letter 2]

19 Aug 2025

Dear Dr Chakrabarti,

We are pleased to inform you that your manuscript 'Inheritable cell-states shape drug-persister correlations and population dynamics in cancer cells' has been provisionally accepted for publication in PLOS Computational Biology.

Best regards,

John P Barton

Guest Editor

PLOS Computational Biology

Mark Alber

Section Editor

PLOS Computational Biology

---

## [Editor Report · Acceptance letter]

PCOMPBIOL-D-25-00043R2

Inheritable cell-states shape drug-persister correlations and population dynamics in cancer cells

Dear Dr Chakrabarti,

I am pleased to inform you that your manuscript has been formally accepted for publication in PLOS Computational Biology. Your manuscript is now with our production department and you will be notified of the publication date in due course.

With kind regards,

Judit Kozma
